# Desmosomal cadherin association with Tctex-1 and cortactin-Arp2/3 drives perijunctional actin polymerization to promote keratinocyte delamination

Oxana Nekrasova[1,2], Robert M. Harmon[1,3], Joshua A. Broussard[1,2], Jennifer L. Koetsier[1], Lisa M. Godsel[1,2], Gillian N. Fitz[1], Margaret L. Gardel[3] & Kathleen J. Green[1,2]

The epidermis is a multi-layered epithelium that serves as a barrier against water loss and environmental insults. Its morphogenesis occurs through a tightly regulated program of biochemical and architectural changes during which basal cells commit to differentiate and move towards the skin's surface. Here, we reveal an unexpected role for the vertebrate cadherin desmoglein 1 (Dsg1) in remodeling the actin cytoskeleton to promote the transit of basal cells into the suprabasal layer through a process of delamination, one mechanism of epidermal stratification. Actin remodeling requires the interaction of Dsg1 with the dynein light chain, Tctex-1 and the actin scaffolding protein, cortactin. We demonstrate that Tctex-1 ensures the correct membrane compartmentalization of Dsg1-containing desmosomes, allowing cortactin/Arp2/3-dependent perijunctional actin polymerization and decreasing tension at E-cadherin junctions to promote keratinocyte delamination. Moreover, Dsg1 is sufficient to enable simple epithelial cells to exit a monolayer to form a second layer, highlighting its morphogenetic potential.

[1] Department of Pathology, Northwestern University Feinberg School of Medicine, Chicago 60611 IL, USA. [2] Department of Dermatology, Northwestern University Feinberg School of Medicine, Chicago 60611 IL, USA. [3] Institute for Biophysical Dynamics, University of Chicago, Chicago 60637 IL, USA. Correspondence and requests for materials should be addressed to K.J.G. (email: kgreen@northwestern.edu)

The epidermis is a dynamic, multilayered epithelium that provides an essential barrier against water loss and environmental insults. The barrier is established through a highly controlled program in which proliferating keratinocytes stop dividing and transit out of the basal layer in a process called stratification. Stratification is coordinated with biochemical and architectural changes necessary to convert cells into a protective outer cornified layer. During this process, the cytoskeleton is re-organized to transform keratinocytes from cuboidal to more flattened shapes as they progress to the upper layers[1]. Studies performed in two-dimensional (2D) keratinocyte cultures suggest that actin remodeling drives changes in gene transcription, as well as cell behavior, to promote differentiation and stratification[2–4]. However, the molecular mechanisms that functionally couple actin reorganization to the initiation of stratification are poorly understood.

Desmosomes are the most abundant adhesive structures in the epidermis[5]. They provide mechanical integrity to the tissue through the anchorage of intermediate filaments (IF) to sites of cell–cell adhesion. Desmosomal cadherins, desmogleins, and desmocollins form the extracellular core of desmosomes and interact with cytoplasmic armadillo proteins, plakophilins, and plakoglobin. Armadillo proteins, in turn, bind desmoplakin (DP), an IF cytolinker[6]. The desmosomal cadherin and armadillo protein families each comprise multiple, differentiation-dependent isoforms. As expression of these isoforms is cell-layer dependent, this leads to differences in desmosome composition during stratification[7,8]. Emerging studies indicate that the regulated expression of desmosomal proteins is vital, not only for epidermal integrity, but also for altering keratinocyte morphology and regulating signaling events that coordinate differentiation and stratification[9,10]. Our laboratory showed that desmoglein 1 (Dsg1), a desmosomal cadherin first expressed as basal cells commit to stratify, regulates keratinocyte morphology as cells transit through the epidermal layers[11]. In particular, suprabasal cells without Dsg1 do not flatten and exhibit large variations in cell size, associated with abnormalities in cytoskeletal architecture.

In a search for associated proteins that could mediate Dsg1-dependent regulation of cell architecture and, therefore promote stratification, we uncovered two binding partners: (1) Tctex-1, a light chain of the dynein motor complex, which targets proteins to dynein during intracellular transport[12]; and (2) cortactin, an actin scaffolding protein, which has previously been shown to promote actin nucleation at E-cadherin-containing cell–cell junctions through recruitment of the Arp2/3 complex[13]. Here we show that initiation of Dsg1 expression in basal cells already adherent through classical cadherins is required for perijunctional actin polymerization, which decreases tension at adherens junctions (AJ), promoting cell transit to the next epidermal layer. Moreover, introducing Dsg1 into simple epithelial cells that do not express this cadherin is sufficient to enable cells to exit from the monolayer to form a second layer. These data provide new insight into how complex epithelia may have arisen during evolution and suggest a mechanism by which Dsg1 promotes stratification through delamination during epidermal morphogenesis.

## Results

**Tctex-1 is a novel binding partner of Dsg1.** We previously showed that Dsg1 silencing impairs differentiated tissue architecture in reconstituted 3D epidermal equivalents. In addition to the reduction of suprabasal keratins and keratohyalin granules, marked changes in cell size and shape were observed[11]. To identify links between Dsg1 and cytoskeleton dynamics that could underlie these observations, we performed a yeast 2-hybrid CytoTrap screen using the Dsg1 cytoplasmic tail (Dsg1-cyto) as bait. Tctex-1, one of the light chains of the cytoplasmic dynein

motor complex[14], which can couple microtubule and actin dynamics[15], was among the positive hits (Supplementary Fig. 1a). Domain mapping indicated that the most C-terminal 140 amino acids of the Dsg1 cytoplasmic tail are sufficient for the interaction (Fig. 1a). Tctex-1 did not interact with another desmoglein isoform, Dsg2 (Supplementary Fig. 1a), and the Tctex-1 binding region in Dsg1 is absent from the other abundant isoform in epidermis, Dsg3, supporting the specificity of the Dsg1–Tctex-1 interaction. Recombinant Dsg1-cytoplasmic tail tagged with GST pulled down endogenous Tctex-1 from keratinocyte lysates, providing further support for the Dsg1–Tctex interaction (Supplementary Fig. 1b).

We next tested whether the Dsg1–Tctex-1 interaction occurs in normal differentiated human epidermal keratinocytes (NHEKs) following the onset of Dsg1 expression after 3 days in 1.2 mM $Ca^{2+}$ containing medium. Tctex exhibited a broadly distributed punctate pattern in the cytoplasm and cell cortex, coming in close proximity with Dsg1 at the cell periphery (Fig. 1b). In epidermal tissue, Tctex-1 was most concentrated in the basal and first suprabasal layer of epidermis, again in a broad pattern of staining in the cell cortex and cytoplasm (Fig. 1c). While areas of close proximity with Dsg1 were observed in both submerged cultures and tissue (Fig. 1b, d, inset), the calculated Pearson's co-efficients for Dsg1–Tctex-1 and E-cad–Tctex-1 did not reveal a strong preference for one over the other.

Therefore, in order to more directly address whether Tctex-1 and Dsg1 exhibit specific and close proximity in situ, we utilized the proximity ligation assay (PLA). PLA signals, visualized as fluorescent spots, serve as a reporter of proteins within 40–100 nm of each other. Dsg1, but not E-cadherin, generated positive PLA signals with Tctex-1 in differentiated NHEKs while silencing of Dsg1 or Tctex-1 inhibited Dsg1–Tctex-1 interactions (Fig. 1e±g; Supplementary Fig 1c–e). We also carried out PLA analysis following introduction of ectopic Dsg1 full length (Dsg1-FL) into undifferentiated cells. Ectopic expression provides a strategy for initiating and synchronizing Dsg1-mediated processes (that are prevented by Dsg1 silencing) by introducing physiologically relevant levels of Dsg1 into undifferentiated cells. This approach also allows a comparison of wild type and mutant versions of Dsg1, without the complicating factor of having to first silence endogenous Dsg1 and provides advantages for optical imaging. Here we compared the ability of Tctex-1 to interact with Dsg1-FL, Dsg2, and E-cadherin. Positive PLA was visualized for FL-Dsg1, but not Dsg2 or E-cadherin, and the signal was abrogated by Tctex-1 knockdown (Fig. 1h–j). We confirmed that Dsg1 knockdown did not affect the level of Tctex-1 and silencing Tctex-1 did not have a measurable effect on the total level of desmosomal or classical cadherin proteins (Fig. 1g, j; Supplementary Fig. 2c). Along with the biochemical analysis, these data support the existence of a Tctex-1 interaction with Dsg1 and not the other cadherins tested.

To determine which Dsg1 domains are important for Tctex-1 binding in NHEKs, we utilized ectopically expressed Dsg1 truncation mutants. Tctex-1 co-precipitated with Dsg1-FL and a Dsg1 construct containing only the cytoplasmic domains (Dsg1-Δ569) with similar efficiency (Fig. 1k, l), indicating that the Dsg1–Tctex-1 interaction does not depend on Dsg1 membrane localization. On the other hand, endogenous Tctex-1 failed to generate a strong PLA signal with ectopically expressed Dsg1 mutants: Dsg1-909 and Dsg1-ICS, lacking the region sufficient to bind Tctex-1 in yeast (Fig. 1k, m, n). While immunofluorescence staining and western blot analysis of the ectopically expressed Dsg1 constructs showed similar expression levels in undifferentiated NHEKs (Supplementary Fig. 1f, g), only Dsg1-FL exhibited a strong PLA signal with endogenous Tctex-1 (Fig. 1m, n).

**Tctex-1 targets Dsg1 to an insoluble membrane pool**. Given that Tctex-1, as a part of the dynein motor complex, is critical for the polarized distribution of membrane proteins such as rhodopsin in simple epithelial cells[12,16], we tested whether Tctex-1 regulates Dsg1 delivery to and/or distribution on the plasma membrane of differentiated keratinocytes. NHEKs were treated with siRNA targeting Tctex-1 and first assayed for endogenous Dsg1 localization after induction of keratinocyte differentiation.

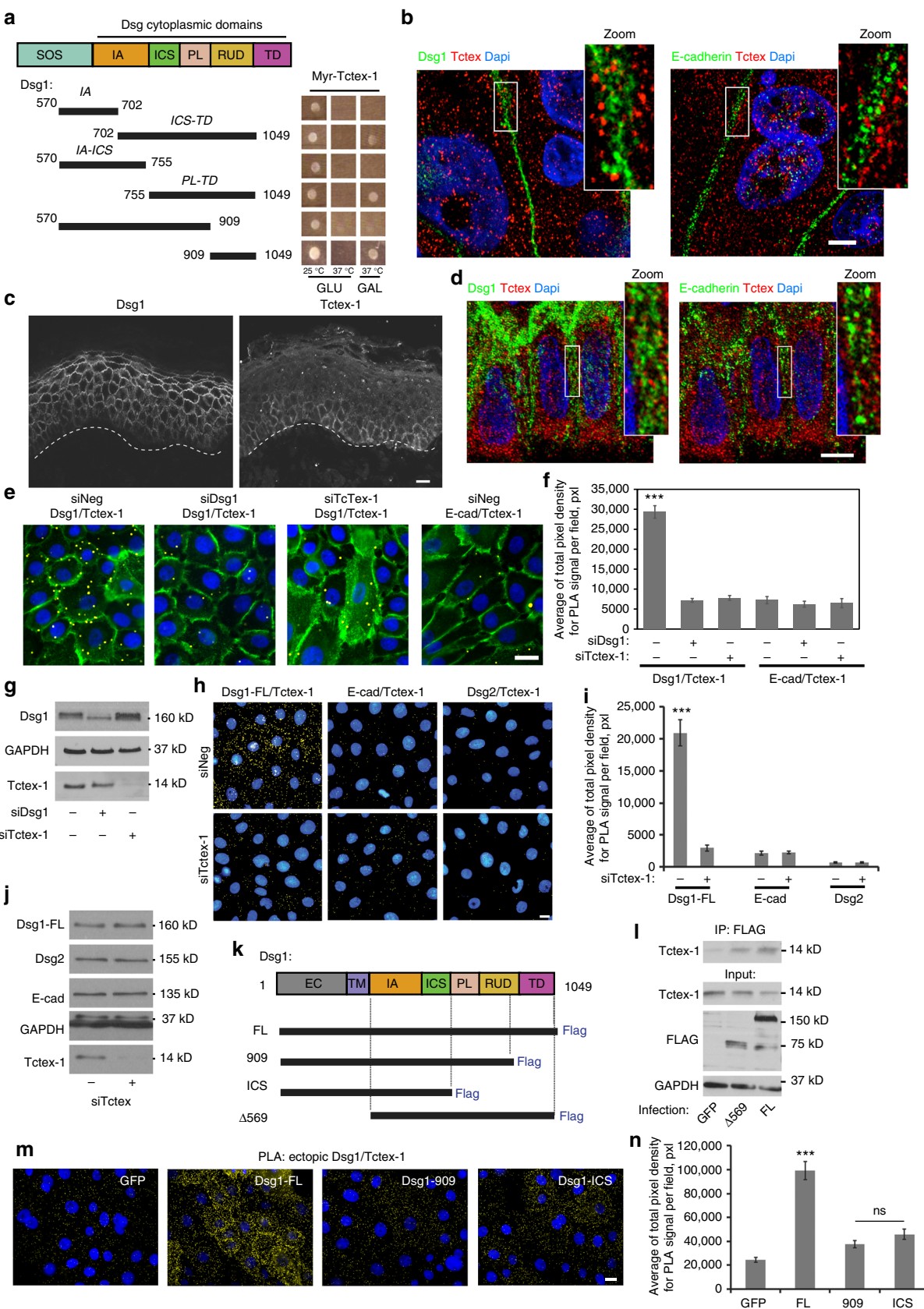

Dsg1 accumulated at cell borders in Tctex-1 knockdown cells to the same extent as it did in control cells transfected with non-targeting siRNA. However, Dsg1 accumulation at cell contacts in the knockdown condition was more broadly distributed and less organized compared with controls (Fig. 2a, b). A similar broadly distributed, less-organized pattern was also observed for the Tctex-1-binding-deficient mutant Dsg1-909 when expressed in undifferentiated NHEKs exposed to 1.2 mM $Ca^{2+}$ for 6 h or 24 h, time points at which endogenous Dsg1 is not yet expressed (Fig. 2c, d). Dsg2 and E-cadherin accumulation and distribution at the cell–cell interface were not detectably altered in Tctex-1-deficient cells, suggesting that, in general, intercellular junctions were maintained (Fig. 2a, b).

To further confirm that Dsg1 can reach the plasma membrane when its interaction with Tctex-1 is abolished, we performed cell surface biotinylation of undifferentiated NHEKs expressing Dsg1-FL or Dsg1-909. Both proteins accumulated at the plasma membrane at similar levels in 1.2 mM $Ca^{2+}$ medium (Supplementary Fig. 2a). Structured illumination microscopy (SIM) imaging of Dsg1-FL or Dsg1-909 revealed that both co-localized with the desmosomal plaque protein DP, an indicator that loss of Tctex-1 binding does not interfere with desmosome assembly (Supplementary Fig. 2b). Moreover, expression of Dsg1-FL or Dsg1-909 increased adhesive strength of epithelial sheets comparably in undifferentiated NHEKs, while silencing of Tctex-1 in differentiated NHEKs did not alter cell–cell adhesion. These observations are consistent with preservation of functional desmosomes (Supplementary Fig. 2c, d). On the other hand, SIM microscopy revealed that abrogation of Dsg1–Tctex-1 interactions caused Dsg1-mCherry-positive desmosome assembly at cell–cell interfaces in an area distinct from endogenous Dsg2-positive desmosomes (Supplementary Fig. 2f). Dsg1 and Dsg2 colocalized with DP in control cultures, with all three proteins overlapping in most areas. While Tctex-1 silencing decreased the overlap between Dsg1 and Dsg2, DP still colocalized with each cadherin. The decrease in Pearson's co-efficient for DP-Dsg is consistent with a constant pool of DP being redistributed among multiple desmosomal cadherins. The data suggest that Tctex-1 is not required for Dsg1-positive desmosome assembly, but instead for the proper position of Dsg1-containing desmosomes at the plasma membrane.

Cell fractionation analysis revealed that Dsg1–Tctex-1 interactions are required to efficiently target Dsg1 to an insoluble biochemical pool distinct from that of E-cadherin. Specifically, while Dsg1-FL distributed equally between the soluble and insoluble membrane fractions, the Dsg1-909 mutant predominantly resided in the soluble membrane fractions, where E-cadherin is concentrated (Fig. 2e, f). A similar shift to the soluble membrane fractions was observed for endogenous Dsg1 in Tctex-1 silenced differentiated keratinocytes compared with control cells treated with non-targeting siRNA (Supplementary Fig. 2e). Furthermore, SIM images showed that the Dsg1-FL localization overlaps with fluorescently labeled ganglioside lipids at cell–cell borders to a greater extent than Dsg1-909 (Fig. 2g, h), which correlates with the increased insolubility observed for Dsg1-FL.

The dynein motor protein is involved in delivery of lipid raft-enriched vesicles to the plasma membrane via either direct exocytosis or during vesicle recycling[17]. Thus, the reduced localization of Dsg1-909 with ganglioside lipids raised the possibility that Tctex-1 may link Dsg1 to the dynein motor to efficiently position Dsg1 at lipid enriched membrane clusters. To address this question, we utilized the Tctex-1 phosphomimetic mutant, T94E, which uncouples Tctex-1 from the dynein complex[15,18]. Since it has been reported that ectopic expression of dynein light chains can suppress endogenous Tctex-1 levels in cells[12], we first checked whether infection with the ectopic Tctex-1 proteins, wild type (Twt) or T94E mutant, has a similar effect in NHEKs. Our data showed that NHEKs have significantly decreased endogenous Tctex-1 when cells are infected with either Twt-Flag or T94E-Flag, which were identified by a molecular weight shift on a western blot (Supplementary Fig. 3a). By immunoprecipitation analysis we determined that the T94E mutation did not prevent binding to Dsg1 (Fig. 2i). Moreover, accumulation of Dsg1-FL at cell borders was not attenuated by expression of Twt or T94E (Supplementary Fig. 3b) and its co-localization with DP suggested that Dsg1-FL still assembles into desmosomes. However, Tctex T94E expression impaired the distribution of Dsg1-FL at borders, in a manner similar to that observed for the Dsg1-909 mutant (Fig. 2c). Furthermore, T94E expression resulted in the partitioning of Dsg1-FL into the soluble membrane pool (Fig. 2j, k), similar to the distribution exhibited by the Tctex-1-binding-deficient mutant Dsg1-909 (Fig. 2e) and endogenous Dsg1 upon Tctex-1 silencing (Supplementary Fig. 2e). Overall, these results indicate that Tctex-1 and its association with dynein are required for targeting of Dsg1 to an insoluble membrane pool where Dsg1 co-localizes more efficiently with Dsg2 at ganglioside-enriched cell–cell interfaces.

**Dsg1–Tctex-1 interactions promote actin reorganization.** Our previous work showed that Dsg1 expression supports

**Fig. 1** Tctex-1 interacts with Dsg1 in keratinocytes. **a** Yeast 2-hybrid (Y2H) analysis of interactions between Sos-tagged Dsg1 and Myr-Tctex-1 protein. Growth on galactose (GAL) at 37 °C indicates an interaction. Incubation on glucose (GLU) at 25 °C represents a permissive growth condition, while incubation at 37 °C on GLU is a control for temperature reversion. IA: intracellular anchor domain, ICS: intracellular cadherin-like sequence domain, PL: intracellular proline-rich linker domain, RUD: repeat unit domain, TD: desmoglein terminal domain. **b** Immunofluorescence analysis of Tctex-1, Dsg1, and E-cadherin localization in differentiating NHEK cells. Scale bar = 5 μm. **c** Immunofluorescence analysis of Dsg1 and Tctex-1 in epidermis. Scale bar = 10 μm. **d** High-resolution analysis of Dsg1, Ecad, and Tctex-1 in the basal layer of the epidermis shows Tctex-1 localization in the cytoplasm and at the cell periphery in close proximity with Dsg1 and E-cad. Scale bar = 5 μm. **e** Proximity ligation assay (PLA) using primary antibodies directed against Dsg1 or E-cad and Tctex-1 performed in differentiated NHEKs with or without Dsg1 or Tctex-1 knockdown. Plakoglobin staining in green marks cell–cell borders. Scale bar = 20 μm. **f** PLA signal area for **e** was counted per field for each condition (>10 cells per field) and averages were determined. Data for one representative experiment out of three biological repeats, $F = 153.581$, ***$p < 0.001$, one way Anova with Tukey test. **g** Western blot showing level of Dsg1 and Tctex-1 knockdown. **h** PLA using primary antibodies directed against Dsg1 and Tctex-1, E-cadherin and Tctex-1, or Dsg2 and Tctex-1 was performed on undifferentiated NHEKs with or without Tctex-1 knockdown. Scale bar = 10 μm. **i** PLA signal area for **h** was counted per field for each condition (>10 cells per field) and averages were determined. Data for one representative experiment out of three biological repeats is shown in the graph (***$p < 0.001$, unpaired two-tailed $t$ test). **j** The level of cadherin proteins and Tctex-1 knockdown is shown on the corresponding western blot. **k** Diagram of Dsg1 Flag-tagged constructs used in the study. **l** Western blot of ectopic Dsg1 Flag-tagged constructs immunoprecipitated from NHEK lysates and probed for Tctex-1. GFP was used as a control. **m** PLA using primary antibody directed against Flag and Tctex-1 was performed on undifferentiated NHEKs retrovirally transduced with GFP, Dsg1-FL-Flag, Dsg1-909-Flag, or Dsg1-ICS-Flag. Scale bar = 10 μm. **n** PLA signal area for **m** was counted per field for each condition (>10 cells per field) (three independent experiments; $F = 47.792$, ***$p < 0.001$, one-way Anova with Tukey test). PLA signal pseudocolored yellow. DAPI staining marks nuclei. Error bars represent standard error of the mean (SEM)

keratinocyte shape transformation during differentiation[11]. Analysis of Tctex-1-deficient differentiated keratinocytes revealed similar alterations in cell size observed previously for Dsg1-deficient cells (Fig. 3a). To determine whether Dsg1–Tctex-1 interactions are required to promote the keratinocyte shape transformation initiated by Dsg1 expression, we analyzed the cell area of undifferentiated NHEKs expressing GFP (as a control), Dsg1-FL or Dsg1-909 after a switch to 1.2 mM Ca$^{2+}$ containing medium. Only NHEKs expressing Dsg1-FL showed a uniform cell area transformation in 2D keratinocyte cultures after 6 h in 1.2 mM Ca$^{2+}$ medium (Fig. 3b). Moreover, Z volume reconstructions revealed that NHEKs with Dsg1–Tctex-1 associations have taller

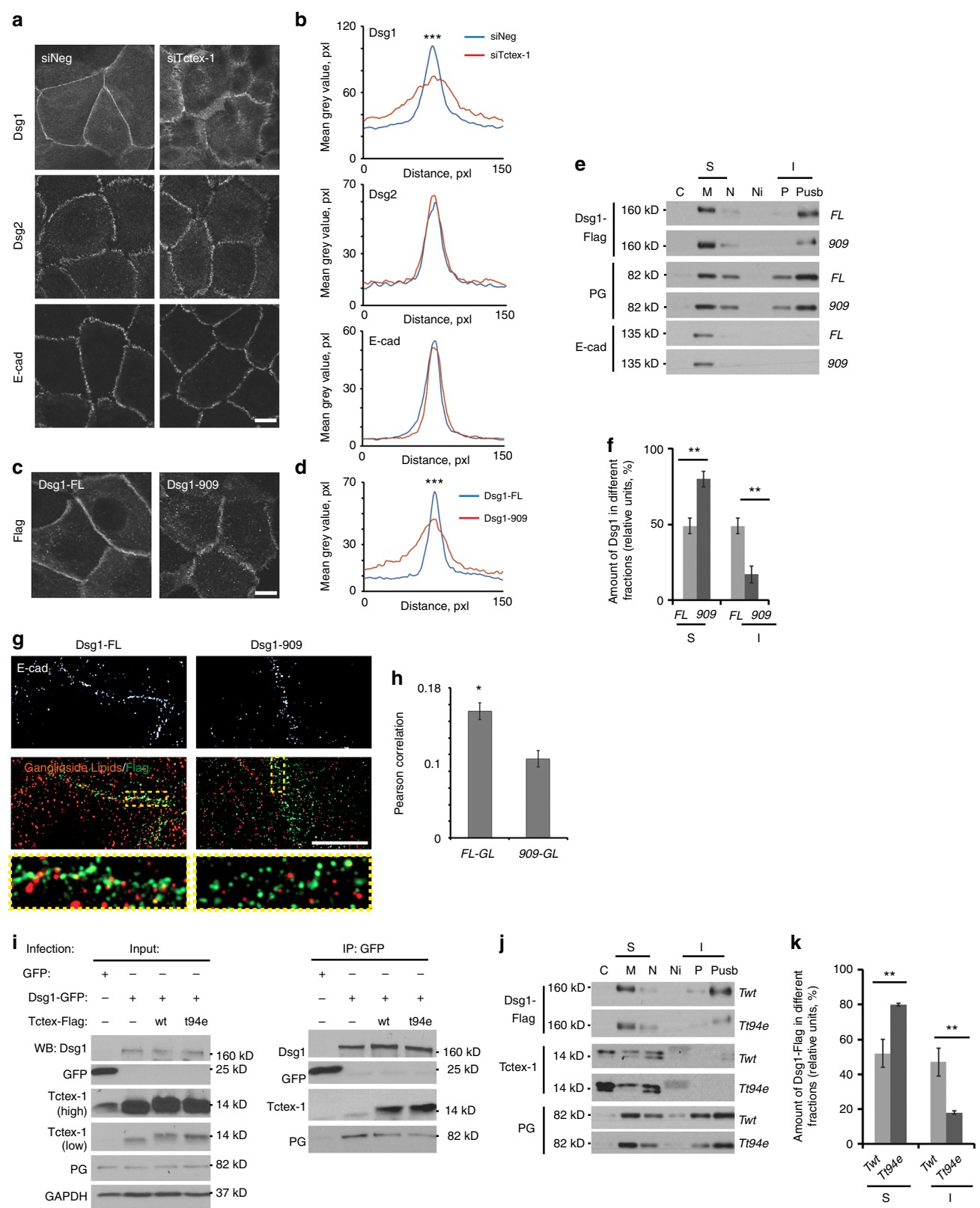

lateral membrane domains, along which Dsg1 does not overlap with E-cadherin-containing adherens junctions (AJs) (Fig. 3c). To determine which cytoskeletal networks might be responsible for these morphological changes, we examined the organization of actin, microtubules and IF in response to Dsg1 expression and association with Tctex-1. Undifferentiated NHEKs were virally transduced with GFP, Dsg1-FL, or Dsg1-909 and switched to 1.2 mM $Ca^{2+}$ medium to trigger junction assembly. After 24 h, cells were stained with phalloidin, anti-tubulin, or anti-keratin antibodies to visualize F-actin, microtubules, or IF, respectively. There were no significant changes in the organization of microtubules or IF at the junctional area (Supplementary Fig. 4a, b). The observation that IF attachments are comparable in Dsg1-FL and Dsg1-909 cells is consistent with results of the dispase assay demonstrating comparable adhesive integrity of substrate detached cell sheets. However, the F-actin network in Dsg1-FL transduced NHEKs differed from that in GFP-control or Dsg1-909 expressing cells. Dsg1-FL positive cells exhibited increased junctional phalloidin staining, characterized by more concentrated, membrane-juxtaposed, cortical actin filaments (Fig. 3d–f). GFP control and the Dsg1-909 mutant failed to elicit similar changes in perijunctional actin and instead exhibited numerous cell-contact-associated radial actin fibers (Fig. 3f). An increased concentration of robust, perijunctional actin filaments was also observed in differentiated keratinocytes (which express endogenous Dsg1) compared with undifferentiated cells and this phenotype was abolished by Tctex-1 knockdown (Supplementary Fig. 4c, d).

The keratinocyte cell shape transformation to tall and cuboidal, from a flat and spread out morphology was previously reported as an outcome of increased perijunctional actin polymerization and a change in actomyosin contractility[4]. Therefore, we first assessed actin incorporation at the cell–cell boundaries by introducing Alexa 568-conjugated-G-actin into lightly permeabilized undifferentiated cells under steady-state conditions. Keratinocytes expressing Dsg1-FL showed a greater number of actin free barbed-ends/de novo nucleation sites, measured by increased G-actin incorporation at the cell cortex compared with GFP-control and Dsg1-909 cells after a 24 h 1.2 mM $Ca^{2+}$ switch (Fig. 3g, h). These data support the idea that Dsg1–Tctex-1 interactions promote actin polymerization at the junctional area.

To determine the role of Dsg1–Tctex-1 interactions in the regulation of actomyosin contractility we measured the impact of Dsg1-FL expression in undifferentiated NHEKs on the cell-matrix traction force distribution within epithelial colonies and compared the results with undifferentiated keratinocytes expressing the Dsg1-909 mutant. The measurement of cell–ECM traction

forces revealed that Dsg1-FL colonies are less contractile than colonies expressing the Dsg1-909 mutant (Supplementary Fig. 5a, b). Moreover, Tctex-1 knockdown in differentiated cells expressing endogenous Dsg1 increased staining of phosphorylated myosin light chain (p-MLC) further confirming an increase in actin contractility at cell–cell interfaces when Dsg1–Tctex-1 interactions are impaired (Supplementary Fig. 5c, d).

The above data suggest that dynein–Tctex-1-dependent Dsg1 positioning at cell junctions reduces actomyosin contractility and promotes robust cortical actin ring formation via junctional actin polymerization.

**Dsg1 recruits cortactin to promote actin polymerization.** Several studies suggest that cortactin is required for cadherin-dependent, Arp2/3-driven actin polymerization at cell–cell interfaces[13,19]. We hypothesized that Dsg1 recruits cortactin–Arp2/3 complexes to desmosomal junctions to promote cortical actin assembly. Indeed, we found that endogenous cortactin co-immunoprecipitated from NHEK lysates with ectopically expressed Dsg1, but not Dsg2 (Fig. 4a). Although Tctex-1 was not required for Dsg1–cortactin interactions (Supplementary Fig. 6a), cell fractionation assays revealed that Dsg1's ability to associate with the Tctex-1–dynein complex is required for the biochemical compartmentalization of cortactin to insoluble membrane fractions (Supplementary Fig. 6b). Furthermore, immunofluorescence analysis of cortactin and Arp3 confirmed their significant increase in junctional localization in undifferentiated keratinocytes expressing Dsg1-FL, but not Dsg1-909 or GFP (Fig. 4b–e) while knockdown of endogenous Dsg1 in differentiated keratinocytes decreased cortactin accumulation at the junctional area (Supplementary Fig. 6c).

Mosaic silencing of cortactin in Dsg1-FL-positive undifferentiated NHEKs prevented perijunctional actin network reorganization (Supplementary Fig. 6d). In addition, Dsg1-FL expressing NHEKs, treated with an Arp2/3-specific inhibitor, CK666, reduced the number of free-barbed ends/de novo nucleation sites at the cell cortex to similar levels as those observed in Dsg1-909 cells (Fig. 4f, g). Together, these results suggest that cortactin–Arp2/3-dependent cortical actin polymerization is occurring at properly localized Dsg1-positive desmosomes.

**Dsg1–Tctex-1 interactions reduce tension at adherens junctions.** Actomyosin contractility plays a crucial role in supporting tension at AJs[20,21]. To test the effect of the observed Dsg1–Tctex-1 dependent decrease in actomyosin contractility on tension

**Fig. 2** Tctex-1 regulates Dsg1 localization at the cell–cell interface. **a** NHEKs treated with control or Tctex-1 siRNA were analyzed for endogenous Dsg1 localization (differentiated condition), or Dsg2 and E-cadherin localization (undifferentiated condition). **b** Line scan analysis of border intensities for Dsg1, Dsg2, and E-cadherin in NHEKs with or without Tctex-1 knockdown (three independent experiments, at least 50 borders were analyzed per condition in representative experiment, ***$p < 0.001$, unpaired two-tailed $t$ test). **c, d** Dsg1-FL-Flag or Dsg1-909-Flag-infected undifferentiated NHEKs were stained for Flag. Line scan analysis of border intensities for ectopic Dsg1 proteins is shown on the graph (four independent experiments, at least 50 borders were analyzed per condition in representative experiment, ***$p < 0.001$, unpaired two-tailed $t$ test). **e** Subcellular fractionation of undifferentiated NHEKs expressing Dsg1-FL-Flag or Dsg1-909-Flag. Collected fractions were analyzed for plakoglobin (PG) (a cytoplasmic constituent of the desmosome), E-cadherin and Flag. C = cytoplasm, S = soluble membrane-bound fractions (M, membrane + N, nuclear), I = insoluble membrane bound fractions (P, pellet + Pusb, urea sample buffer solubilized pellet). **f** Quantification of the percentage of Dsg1-FL-Flag or Dsg1-909-Flag in soluble and insoluble membrane fractions (five independent experiments; **$p < 0.01$, unpaired two-tailed $t$ test). **g** Structured illumination microscopy images of ectopic Dsg1-FL-Flag or -909-Flag mutant (green) and CT-B-labeled lipid rafts (red). E-cadherin staining in the top panels indicates the cell–cell border location. Rectangles mark zoomed in areas. **h** Pearson's correlation coefficients for Dsg1-FL-Flag/CT-B-labeled lipid rafts or Dsg1-909-Flag/CT-B-labeled lipid rafts (from one representative experiment out of 3, *$p < 0.05$, unpaired two-tailed $t$ test). **i** Western blot of GFP immunoprecipitation from undifferentiated NHEKs expressing GFP or Dsg1-GFP$^{+/-}$ ectopic Tctex constructs probed for Dsg1, Tctex-1, GFP, and PG (positive control for Dsg1 interactions). **j** Subcellular fractionation of undifferentiated NHEKs co-expressing Dsg1-FL-Flag with Tctex-wt-Flag (Twt) or Tctex-t94e-Flag (Tt94e). Collected fractions were analyzed for PG, Dsg1, and Tctex-1. **k** Quantification of the percentage of Dsg1-FL-Flag in soluble and insoluble fractions in the presence of Tctex-wt or Tctex-t94e (three independent experiments; **$p < 0.01$, unpaired two-tailed $t$ test). Scale bars = 10 μm. Error bars represent SEM

distribution at AJs, we first analyzed junctional staining of endogenous vinculin, which has been shown to accumulate at AJs in response to tension[22,23]. Ectopic expression of Dsg1-FL, but not the Dsg1-909 mutant in undifferentiated keratinocytes, reduced the fluorescence intensity of vinculin at cell junctions (Fig. 5a, c) and decreased the area of cell junctions occupied by vinculin (Fig. 5b, d). This implies that tension at AJs was reduced

in Dsg1-FL positive cells. Next, we utilized a Förster resonance energy transfer (FRET)-based E-cadherin tension sensor (E-cad-TSmod) to detect tensile stretch of the E-cadherin cytoplasmic tail[24]. For this sensor, increasing tension results in decreasing FRET as the donor and acceptor fluorophores are pulled further apart. When expressed in undifferentiated NHEKs, E-cad-TSmod accumulated at the cell–cell contacts during the Ca[2+] switch time

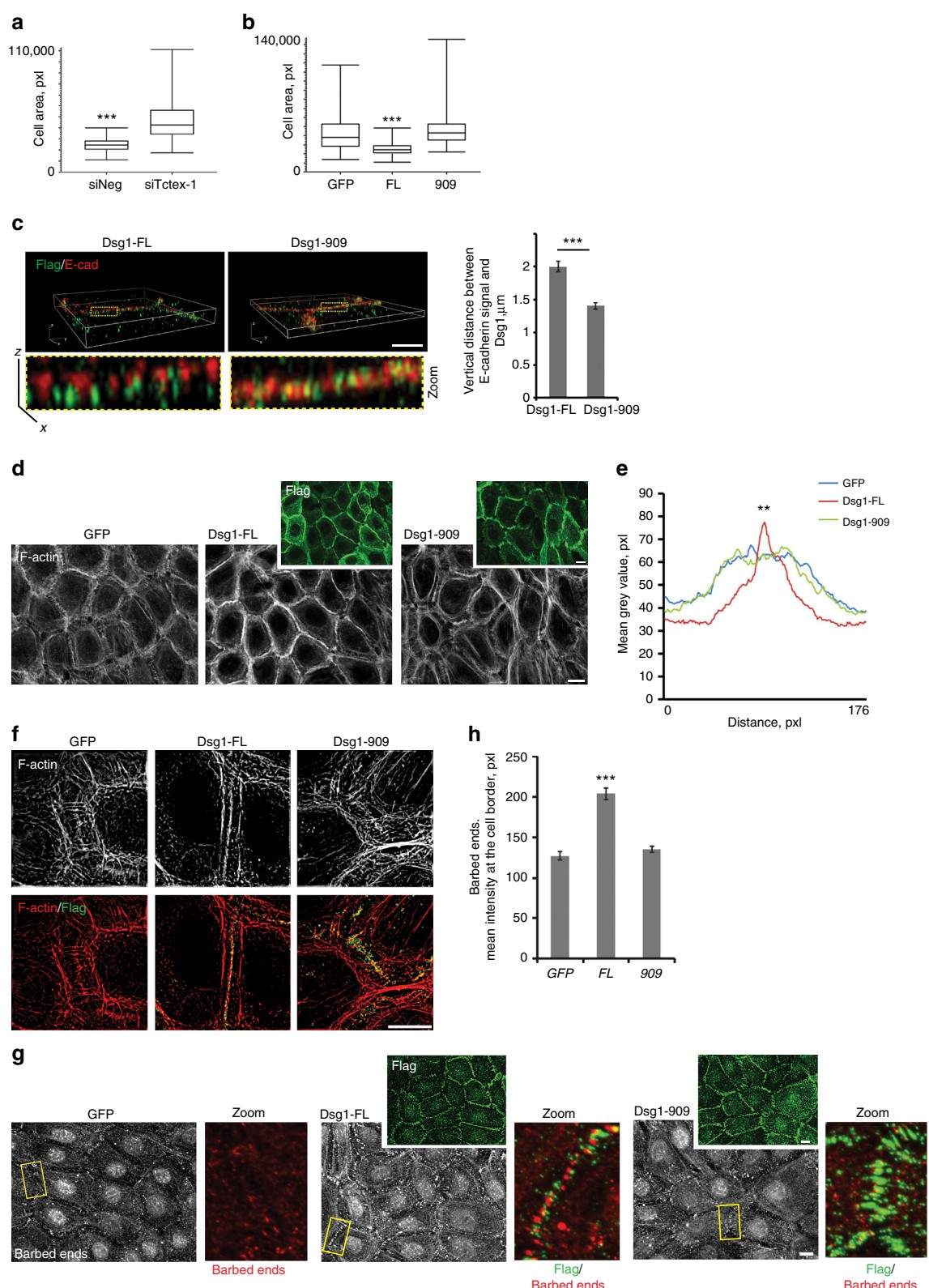

course (1 h, 6 h, and 24 h), in a manner similar to endogenous E-cadherin[25]. The junctional E-cad-TSmod FRET signal decreased during the Ca$^{2+}$ switch time course (Supplementary Fig. 7a, b), consistent with increased E-cadherin tension during cell–cell contact maturation[24,26]. Coexpression of ectopic Dsg1-FL increased the FRET index of E-cad-TSmod after 24 h in 1.2 mM Ca$^{2+}$ medium at cell contacts and at the contact-free plasma membrane area, signifying reduced tension on plasma membrane bound E-cadherin regardless of whether E-cadherin was recruited to cell–cell contacts (Fig. 5e, f). In contrast, coexpression of mCherry or the Dsg1-909 mutant had no effect on E-cad-TSmod FRET. Furthermore, knockdown of Tctex-1 in undifferentiated NHEKs expressing Dsg1-FL decreased E-cad-TSmod FRET back to control levels (Fig. 5g, h).

Finally, we used two-photon laser ablation to measure changes in contractile tension at cell–cell junctions when Dsg1-FL is expressed in undifferentiated keratinocytes. Initial recoil of E-cadherin-labeled cell–cell junctions from the cut site was significantly slower in cells with Dsg1-FL compared to mCherry control or Dsg1-909-mutant expressing cells, signifying a decrease in contractile tension (Fig. 6a–d; Supplementary Movies 1-3).

Altogether these data strongly suggest that Tctex-1-dependent Dsg1 insoluble plasma membrane compartmentalization at the cell–cell interface, followed by Arp2/3-dependent actin polymerization, affects AJ tension in NHEKs.

**Dsg1–Tctex-1 interactions promote keratinocyte delamination**. It has been shown that cell extrusion in simple epithelia requires decreased contractility at the apical cell–cell interface of the cell targeted for extrusion[27,28]. To test whether Dsg1 expression and proper plasma membrane compartmentalization can promote cell extrusion from simple epithelia that do not express endogenous Dsg1, we utilized MDCK cells as a model. MDCK cells, which express endogenous Tctex-1 and cortactin, were transduced with constructs for expression of Dsg1-FL or Dsg1-909 and monolayer behavior was recorded overnight. MDCK cells transfected with Dsg1-FL exhibited dramatically increased numbers of extrusion events from the monolayer compared to GFP expressing control cells (Supplementary Movies 4 and 5). Furthermore, extruded Dsg1-expressing cells remained attached to the cell monolayer, resulting in focal regions of stratification. MDCK cells expressing the Dsg1-909 behaved similarly to the GFP control and did not develop these stratified regions (Fig. 7a, b). Moreover, knockdown of Tctex-1 in MDCK cells expressing Dsg1-FL decreased the extrusion events to the level of Dsg1-909 expressing cells and GFP control, suggesting that Dsg1–Tctex-1 interactions are required for promoting apical extrusion from the monolayer (Fig. 7c).

Since cell extrusion in simple epithelia can be increased due to apoptotic stimuli within the monolayer to repair the barrier or when epithelial cells become too crowded due to an increase in proliferation[29], we next tested whether Dsg1-FL expression promoted extrusion independent of apoptosis stimuli or an increase in cell division. First, we performed BrdU pulse labeling in Dsg1-FL, Dsg1-909, or GFP-positive monolayers to quantify mitotic cells. No difference was detected using either the BrdU incorporation assay or by counting Ki67-positive cells as a measure of proliferation (Supplementary Fig. 8a,b). Next, we analyzed the rate of apoptosis within the monolayer of GFP, Dsg1-FL, or Dsg1-909 positive cells utilizing the TUNEL assay and by immunoblotting for Bcl-2 and cleaved Caspase 3 in total cell lysates. These tests did not reveal any indication of increased apoptosis in Dsg1-FL expressing monolayer compared to the GFP-control and Dsg1-909 expressing monolayer (Supplementary Fig. 8c, d). When taken together, our data are consistent with the idea that Dsg1 expression promotes cell extrusion from the simple epithelial monolayer due to decreased tension at AJs and Arp2/3-mediated actin polymerization at the cell cortex.

Two major processes contribute to stratification leading to the formation of the multi-layered epidermis during various stages of development and differentiation: asymmetric divisions dictated by changes in spindle pole orientation, and delamination resulting from single cell detachment from the basement membrane and transit out of the monolayer to form a new layer[30]. To test whether Dsg1 can promote formation of a multi-layered epithelium in NHEKs, we adapted a system used to quantify stratification events in the developing mouse epidermis[31]. We utilized 3D epidermal organotypic cultures containing less than 10% GFP-labeled keratinocytes. The sparse distribution of GFP cells allowed us to analyze cell behavior during the first 24 h of stratification, categorizing them into four groups. First, a solitary GFP-labeled cell located in the integrin β4 positive basal cell layer was scored as a single basal cell event. Second, the observation of two neighboring GFP-labeled cells in direct contact and located in the integrin β4 positive basal layer was interpreted as evidence of a symmetric cell division event. Third, two neighboring GFP-labeled cells in direct contact with one cell in the integrin β4 positive basal layer and one directly above it was scored as an asymmetric cell division event. Finally, the observation of a single GFP-labeled cell located above the integrin β4 positive cell layer was considered to have undergone delamination (Fig. 8a, b). To set a baseline for this assay, we carried out a population analysis at day 0 (D0), which is 48 h after seeding of the cells onto the collagen plug and before exposing cells to the air–liquid interface to induce formation of a multi-layered epithelium. We scored 21 ± 2% of GFP-labeled cells as

**Fig. 3** Dsg1–Tctex-1 interactions promote perijunctional actin polymerization. Quantification of the cell area of differentiated NHEKs treated with control (siNeg) or Tctex-1 siRNA (**a**) or undifferentiated NHEKs infected with GFP, Dsg1-FL-Flag, or Dsg1-909-Flag (**b**) shown in Whisker plots. Box boundaries indicate the range of cell areas measured, middle bars depict the mean of compiled data sets, whiskers represent the maximum and the minimum of the measured areas (collected from three independent experiments and areas of at least 125 cells were measured per condition, ***$p < 0.001$, unpaired two-tailed $t$ test in **a** and one-way Anova with Tukey test in **b**, $F = 68.519$). **c** Z-volume reconstructions of the cell–cell borders of undifferentiated NHEKs retrovirally transduced with Dsg1-FL-Flag or Dsg1-909-Flag. NHEKs were stained for Flag (green) and E-cadherin (red) and imaged using structured illumination microscopy (SIM). Rectangles mark zoomed in areas. The lateral distance between E-cadherin and Dsg1 at cell contacts of NHEKs is quantified to the right (three independent experiments; ***$p < 0.001$, unpaired two-tailed $t$ test). **d** GFP, Dsg1-FL-Flag, or Dsg1-909-Flag-infected NHEKs were stained for F-actin and Flag (green). **e** Line scan analyses of cortical F-actin intensities at the cell–cell interface in undifferentiated NHEKs expressing GFP or ectopic Dsg1-Flag constructs (three independent experiments, at least 50 borders were analyzed per condition in each experiment ($F = 46.432$, **$p < 0.01$, one-way Anova with Tukey test). **f** Representative SIM images of F-actin (gray/red) and Flag (green) at the cell–cell interface of undifferentiated NHEKs expressing GFP, Dsg1-FL-Flag, or Dsg1-909-Flag. **g** Barbed-end labeling (G-actin incorporation) at cell junctions in undifferentiated NHEKs expressing GFP, Dsg1-FL-Flag, or Dsg1-909-Flag (Flag staining shown in top insets, green). Rectangles mark zoomed junctional regions, which are shown to the right, barbed ends (red) and Flag (green). **h** Quantification of barbed-end labeling in undifferentiated NHEKs expressing GFP or ectopic Dsg1 constructs (at least 40 borders per condition were analyzed from each experiment, three independent experiments; ***$p < 0.001$, one-way Anova with Tukey test). Scale bars = 10 μm. Error bars represent SEM

delamination events and 56 ± 1% as single basal cell events (mean ± SEM (standard error of the mean), three biological repeats). At 24 h after being exposed to the air–liquid interface (day 1 (D1)), the number of basal cell events decreased to 35 ± 3%, while the number of delamination events scored significantly increased up to 44 ± 2% (out of five biological repeats), suggesting that basal cells had delaminated (Fig. 8c). Furthermore, silencing either Dsg1 or Tctex-1 decreased the number of delamination events

and significantly increased the number of GFP-labeled single cells present in the integrin β4 positive basal cell layer, while knockdown of Dsg2 did not have a significant effect on GFP-labeled cell distribution (Fig. 8d–f; Supplementary Fig. 10a).

Since changes in the actin cytoskeleton and cell shape are required for efficient spindle re-orientation in keratinocytes[32,33], we also assessed the number of symmetric and asymmetric events in 3D epidermal organotypic cultures under control, Dsg1, Tctex-

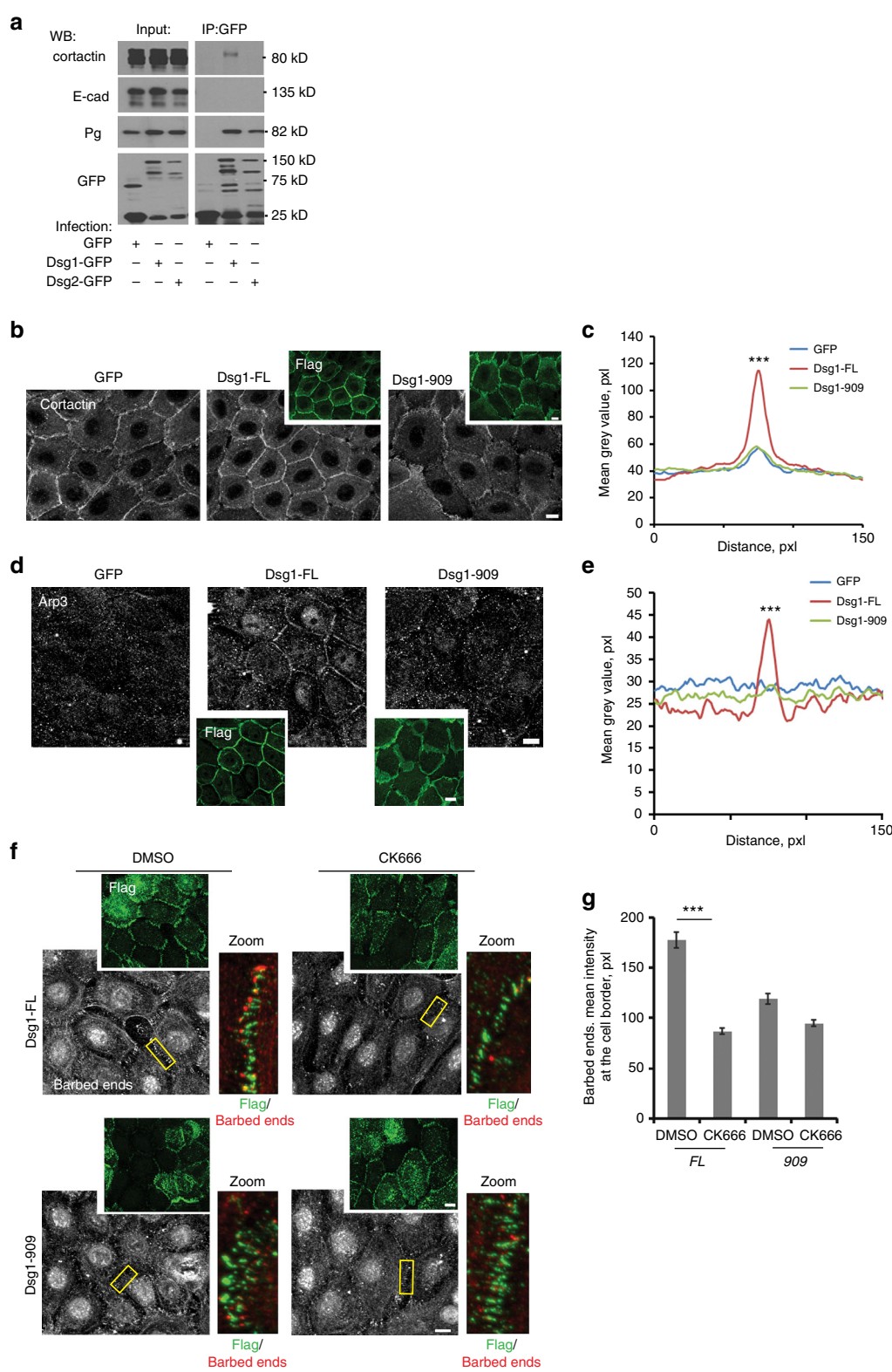

1, or Dsg2 knockdown conditions. Silencing Dsg1, Tctex-1, or Dsg2 did not significantly affect the number of symmetric or asymmetric division events (Fig. 8d–f; Supplementary Fig. 10a). However, because the number of division events in this assay was small, we also stained for γ-tubulin as a marker of the centrosome to check for defects in cell polarity, and survivin, a cleavage furrow marker, to identify epidermal division angles. Knockdown of Dsg1 or Tctex-1 did not affect the apical position of centrosomes relative to the nucleus in basal keratinocytes (Supplementary Fig. 9a, b). Moreover, the analysis of epidermal division angles showed that a large majority of cell divisions in control and knockdown D1 organotypic raft cultures occurred with a spindle oriented parallel to the basement membrane (Supplementary Fig. 9c, d), which supports our population analysis of D1 rafts (Fig. 8c–f) and corresponds with previously observed data at E12.5–13.5 in mouse embryos[31]. Altogether, these results suggest that decreased tension at the AJs due to Dsg1 expression and its proper positioning at the lateral cell interface promotes basal cell delamination and does not affect asymmetric cell division. However, we do not rule out the possibility that Dsg1 could have an impact on asymmetric divisions in certain contexts.

It has been observed that prior to the formation of a second layer, the single-layered, immature ectoderm morphologically resembles a simple polarized epithelium[34]. We hypothesized that the initial expression of Dsg1 in polarized immature ectoderm might decrease tension at the established E-cadherin junctions to allow cell delamination. To determine the plausibility of this idea, we examined Dsg1 expression in developing murine epidermis. At E9 Dsg1 expression cannot be detected in the ectodermal monolayer. However, at E11, when formation of a superficial temporary protective layer called the periderm begins[34,35], Dsg1 can be observed in groups of basal cells beneath periderm cells and in periderm cells (Fig. 8g; Supplementary Fig. 10b). Western blot analysis of E9, E11, and E14 embryos confirmed that Dsg1 is present at E11, when keratin 17, the first marker of periderm, is detected (Fig. 8h). These observations are consistent with a possible role for Dsg1 in redistributing membrane tension during periderm formation.

## Discussion

Changes in the distribution of tension along apposing membranes mediated by AJs plays an important role in governing the behavior of cell sheets and tissues during development and tissue remodeling[24,27,36,37]. While studies in 2D keratinocyte cultures show that initial steps of stratification and differentiation are tightly correlated with remodeling of the actin cytoskeleton, it remains unclear how tension patterns are regulated during multilayer epidermal tissue formation. Our work reveals that the onset of expression of the differentiation-dependent cadherin Dsg1 in keratinocytes, which are already adherent through the evolutionarily more primitive adhesive machinery including E-

cadherin and basal desmosomal cadherins, causes actin re-organization and changes the homeostatic contractile tension associated with AJs of neighboring cells. The reduced tension at AJs is associated with exit of cells from the basal layer in newly stratifying epidermal cultures. Importantly, introducing Dsg1 into simple epithelial cells that do not express this cadherin but do have the baseline machinery required for actin reorganization is sufficient to promote extrusion events to establish a multi-layered tissue structure.

A prerequisite for the observed actin re-organization is proper membrane compartmentalization of Dsg1, which we demonstrated depends on a newly identified binding partner, the dynein light chain, Tctex-1. Tctex-1 is a cargo adaptor for dynein motor transport, and has been shown to be required for the polarized delivery of the rhodopsin receptor to the apical surface of epithelial cells[12,16]. Here, we demonstrated that Tctex-1, as a part of the dynein complex, is required for localization of Dsg1-containing desmosomes in highly insoluble membrane domains at the cell–cell interface, enriched with ganglioside lipids and co-localizing with Dsg2. Tctex-1-dependent Dsg1 insoluble membrane compartmentalization recruits the actin scaffolding protein, cortactin, to the same locations providing a platform for desmosomal cadherin-actin cooperation.

It was previously determined that cortactin associates with classical cadherins at the epithelial zonula adherens (ZA), a continuous zone of specialized adherens junctions in simple polarized epithelial cells[13,19]. Recruitment of cortactin to the ZA promotes Arp2/3-dependent actin reorganization[13], which is further stabilized into perijunctional bundles by the cortactin binding protein N-WASP. These actin bundles in turn stabilize cell–cell contacts, ultimately providing junctional tension at the ZA needed for simple epithelial tissue integrity and morphogenesis[27,38,39]. When Arp2/3-dependent actin nucleation is not followed by N-WASP stabilization, apical tension at cell–cell interfaces is redistributed leading to cell extrusion from the monolayer[27]. However, the potential for alteration of tension at AJs during epidermal stratification has not been fully investigated. A recent report showed that mouse keratinocytes with a loss of the ArpC3 subunit of Arp2/3 produced robust F-actin bundles compatible with increased AJ associated contractility[40]. Utilizing vinculin junctional staining as a marker for AJ associated tension, and an E-cadherin FRET sensor, we revealed that proper positioning of the Dsg1 at the cell–cell interface reduced tension at the AJ. These observations were further confirmed by measuring initial recoil at the cell junctional area in cells expressing Dsg1. Consistent with this model, we demonstrated that disruption of Dsg1 and Tctex-1 association impaired Dsg1's ability to reduce cortical tension, corresponding with decreased keratinocyte delamination from monolayers undergoing the initial stages of stratification.

We previously showed that Dsg1 expression is required for keratinocyte progression through a terminal differentiation program[11,41]. At the same time, Dsg1 increases intercellular adhesion

---

**Fig. 4** Dsg1 recruits cortactin–Arp2/3 complexes to junctions to promote cortical actin polymerization. **a** Western blot of GFP immunoprecipitations from undifferentiated NHEKs expressing GFP, Dsg1-GFP or Dsg2-GFP probed for GFP, cortactin, Pg (positive control for Dsg1/Dsg2 interactions), and E-cadherin (negative control). **b–e** GFP, Dsg1-FL-Flag, or Dsg1-909-Flag-infected undifferentiated NHEKs were stained for cortactin and Flag (staining for Flag shown in top insets) (**b**) or Arp3 and Flag (**d**). Line scan analyses of cortactin intensity (**c**) or Arp3 intensity (**e**) at the cell–cell interfaces in undifferentiated NHEKs expressing GFP, Dsg1-FL-Flag, or Dsg1-909-Flag (at least 50 borders per condition were analyzed in each experiment, three independent experiments, ***$p < 0.001$, one-way Anova with Tukey test; $F = 13.703$ for **c** and $F = 69.212$ for **e**). **f** Barbed-end labeling at cell junctions of undifferentiated NHEKs expressing ectopic Dsg1-FL-Flag or Dsg1-909-Flag treated with DMSO or CK666. Flag staining is shown for each condition (insets, green). Rectangles mark zoomed in junctional regions, which are to the right of each panel, barbed ends (red) and Flag (green). **g** Quantification of barbed-end labeling in undifferentiated NHEKs expressing Dsg1-FL-Flag (FL) or Dsg1-909-Flag (909) treated with DMSO or CK666 (at least 60 borders were measured per condition from each experiment, three independent experiments; ***$p < 0.001$, unpaired two-tailed $t$ test). Scale bars = 10 μm. Error bars represent SEM

strength, through its association with the keratin IF network (Supplementary Fig. 2d). The work presented here reveals that Dsg1 also decreases junctional tension associated with AJs, resulting in delamination of Dsg1-expressing cells from the basal layer of newly stratifying epithelial cell cultures. We propose a model whereby the onset of expression and correct positioning of Dsg1 helps coordinate three distinct processes: a local increase in cell–cell adhesion, a biochemical program of terminal differentiation and remodeling of cytoplasmic and junctional actin to promote basal cell delamination. This latter actin remodeling process integrates the loss of orthogonally oriented cell-contact-associated F-actin bundles, which are exerting a pulling force on E-cadherin, with cortactin–Arp2/3-dependent cortical actin polymerization at Dsg1-positive desmosomes. Changes in both

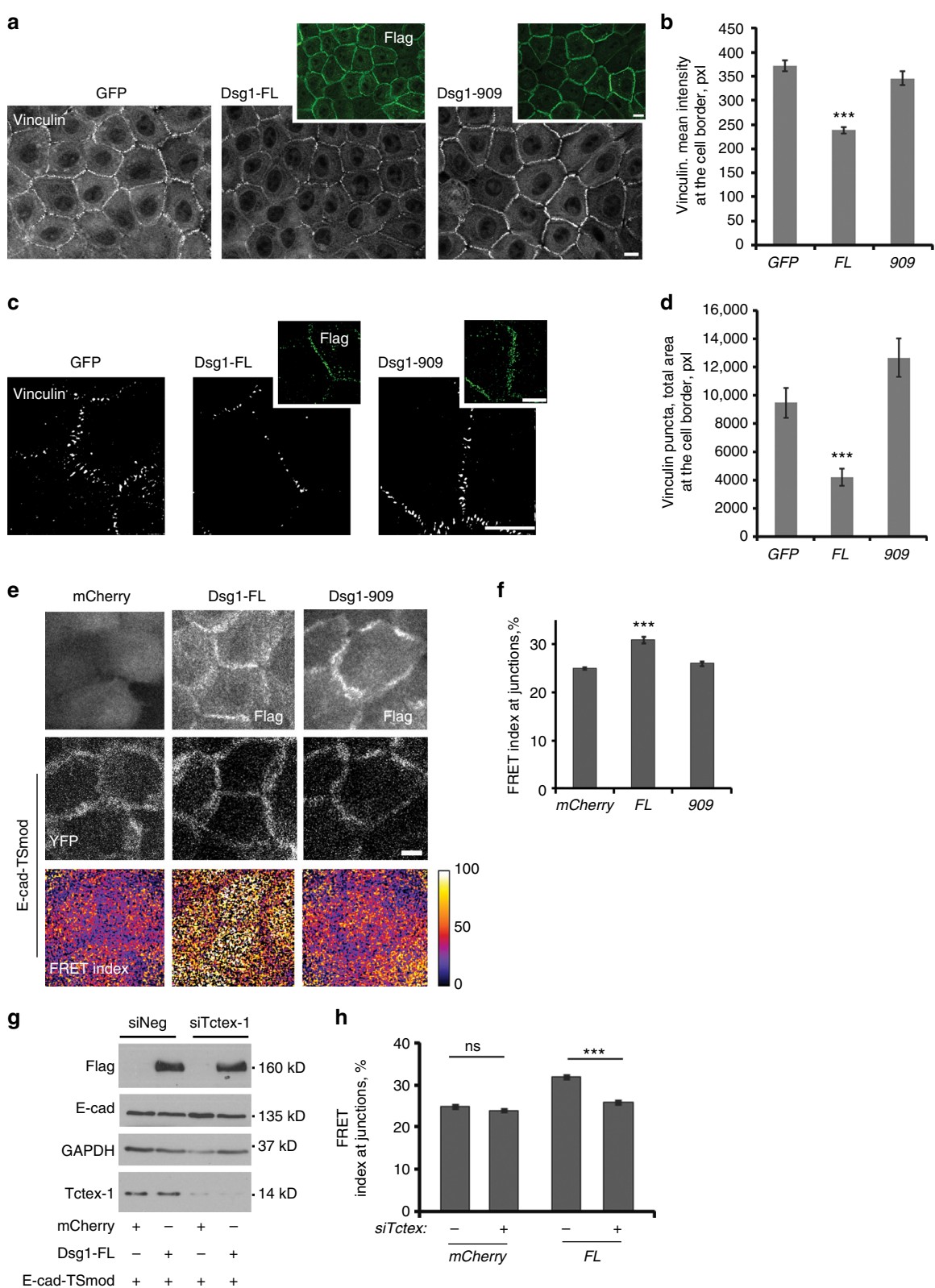

actin populations at the cell–cell interface are likely to contribute to the observed decrease in tension at junctions that consequently promote basal cell delamination during the initial steps of stratification (Fig. 9).

Our data provide a new perspective on a report from Wickstrom and colleagues, which was published while this work was in review. They propose a model whereby proliferation in the epidermal basal layer results in crowding, resulting in local alterations in cell shape and stress distribution. These changes are associated with decreased cortical tension and increased cell–cell adhesion necessary for cells to delaminate[42]. Our data show that Dsg1 plays a central role in this process, orchestrating changes in tension and adhesions while ensuring the mechanical integrity of the stratifying epithelial sheet through connections with the IF cytoskeleton. The Tctex-1 deficient phenotype naturally sorts out these two modes and uses of adhesion machinery. Overall, our work highlights the importance of Dsg1 in the morphogenesis of complex multi-layered epithelia.

## Methods

**Cell culture**. Primary normal human epidermal keratinocytes (NHEKs) were isolated from human foreskin as previously described[43] and grown in M154 media supplemented with 0.07 mM CaCl$_2$, human keratinocyte growth supplement (HKGS) and gentamicin/amphotericin B solution (Thermo Fisher Scientific). Keratinocytes were transduced with retroviral supernatants produced from Phoenix cells (provided by G. Nolan, Stanford University, Stanford CA) as previously described[44]. Briefly, Phoenix cells transiently transfected with retroviral cDNA constructs were harvested at 70% confluency for 12–24 h at 32 °C. Supernatants were collected and concentrated using Centricon Plus-20 columns (EMD-Millipore). Infection of keratinocytes was done at 15% cell confluency; cells were incubated at 32 °C for 1 h in M154 media containing 4 µg/ml polybrene hexadimethrine bromide (Sigma-Aldrich) and retrovirus supernatants.

For data generated from submerged cultures, keratinocytes were grown to confluency and switched to M154 media supplemented with HKGS, gentamicin/amphotericin B and 1.2 mM CaCl$_2$ for either 6 h or 24 h (for transduced cells) or for 3 days (for cells without retroviral transduction to induce junction formation and cell differentiation). To perform transient knockdown with siRNA, keratinocytes were electroporated using the Amaxa Nucleofector System (Lonza) according to manufacturer's instructions. Briefly, keratinocytes were counted and re-suspended in Ingenio Electroporation Solution (Mirus) and siRNA (final concentration 20 µM) and then electroporated using program X-001.

For stratified organotypic epidermal raft cultures, keratinocytes were expanded and grown at an air-medium interface according to published protocols[45]. Organotypic epidermal raft cultures were grown for 24 h at an air-medium interface and then were lysed in Urea Sample Buffer (8 M deionized urea; 1% SDS; 10% glycerol; 60 mM Tris, pH6.8; 0.1% pyronin-Y; and 5% β-mercaptoethanol) for biochemical analysis or fixed in 4% formalin solution for 3 h at 4 °C followed by an incubation in 15% sucrose overnight at 4 °C and subsequent embedding of samples in Optimal Cutting Temperature (O.C.T.) Compound (Tissue Tek) for histological analysis.

MDCK cells (gift from Dr. P. Kopp, Northwestern University) were maintained in minimum essential medium (MEM) supplemented with 5% FBS, 100 U/ml penicillin, and 100 U/ml streptomycin (Mediatech). All experiments were performed in DME media supplemented with 10% FBS, 100 U/ml penicillin, and 100 U/ml streptomycin. To perform transient knockdown with siRNA, MDCK cells were electroporated using the Amaxa Nucleofector System (Lonza) according to manufacturer's instructions. Retroviral infection of MDCK cells was done as described above for keratinocytes. To polarize cells, 750,000 MDCK cells were grown on 24 mm Transwell inserts (0.4 µm pore, Costar) for 4 days.

**DNA constructs and siRNA**. LZRS-GFP, LZRS-FLAG Dsg1(FL), LZRS-FLAG Dsg1 (Δ569), LZRS-shLmn, and LZRS-shDsg1 were generated and described by Getsios et al.[11]. LZRS-GFP-FLAG Dsg2 was generated by Chen et al.[46]. GST-Dsg1 (cyto), Sos-Dsg1(cyto), Myr-PG, Sos-Dsg1(IA), Sos-Dsg1(ICS), and LZRS-FLAG Dsg1(ICS) were generated and described by Harmon et al.[41].

Sos-Dsg1(PL-TD), Sos-Dsg1(570–909), and Sos-Dsg1(909–1049) were generated by ligation of PCR products encoding nucleotides 2478–3360, 1923–2940, and 2940–3360 of sequence from Genbank: NM-001942.3 into pSos (Agilent Technologies) respectively. Sos-Dsg2 (cyto) was generated by ligation of PCR product encoding nucleotides 2154–3603 of sequence from Genbank: NM-001943.4 into pSos (Agilent Technologies). Myr-Tctex-1 was generated by ligation of PCR product encoding nucleotides 8–403 of sequence from Genbank: NM_006519.2 into pMyr (Agilent Technologies). The FLAG Dsg1 (FL) fragment encoding nucleotides 213–3360 of sequence from Genbank: NM-001942.3 was amplified by PCR and cloned into the pEGFP-N2 vector. The resulting GFP-FLAG Dsg1 fragment was then cloned into the pLZRS (provided by M. Denning, Loyola University, Chicago, IL). LZRS-FLAG Dsg1(909) was generated by ligation of PCR product encoding nucleotides 213–2940 of sequence from Genbank: NM-001942.3 into pLZRS. LZRS-FLAG Tctex-1 was generated by ligation of PCR product encoding nucleotides 8–403 of sequence from Genbank: NM_006519.2 into pLZRS. The LZRS-FLAG Tctex-1 T94E mutation was introduced into the LZRS-FLAG Tctex-1 backbone using a site-directed mutagenesis kit (QuikChange; Agilent Technologies). E-cadherin-TSMod, E-cadherin-TSmod-mTFP, and E-cadherin-TSMod-eYFP were a gift from Alexander Dunn (Stanford University). NotI was used to remove the coding sequence from these plasmids and to insert them into pLZRS.

Non-targeting (NT) siRNA (negative control) (Dharmacon), siRNA against human Tctex-1 (Integrated DNA Technologies) with targeting sequences: 5′-CCA AATGACCGCACTGTGATGTGAA-3′ and 5′-ATAGAAAGCGCAATTGGTGG TAACG-3′, siRNA against canine Tctex-1 (Sigma-Aldrich) with targeting sequence: 5′-ACAAATGTCGTAGAACAAACT-3′, siRNA against Dsg1 (Integrated DNA Technologies) with targeting sequence: 5′-CCATTAGAGAGTG GCAATAGGATGA-3′, siRNA against Dsg2 (Integrated DNA Technologies) with targeting sequence: 5′-CCTGGAAGCAGAGACAGTGTGGTCCTT-3′ and siRNA against cortactin (Integrated DNA Technologies) with targeting sequences: 5′-GCT TTACCACAATGAGCAATGAGGT-3′ and 5′- CAAGCTTCGAGAGAATGTC TT-3′ were used for transient knockdown in keratinocytes.

**Antibodies**. The following primary antibodies were used: rabbit Rb-5 anti-Dsg2 (Progen Biotechnik, catalog # 610121, 1:1000 WB, 1:100 IF), goat anti-Dsg1 (R&D Systems, catalog # AF944, 1:1000 WB, 1:100 IF), human 982 anti-Dsg1 (pemphigus foliaceus sera, 1:300 IF), mouse P124 anti-Dsg1 (Progen Biotechnik, cat#651111, 1:25 IF), rabbit anti-Dsg1 (Abcam, catalog# ab124798, 1:200 IF), mouse anti-DYNLT1(Tctex-1) (EMD-Millipore, cat#MAB1076-I, 1:500 WB), mouse anti-DYNLT1 (Tctex-1) (Sigma-Aldrich, catalog# D9944, 1:100 IF, 1:1000 WB), mouse anti-Tctex-1 (gift from K. Pfister, University of Virginia School of Medicine, 1:100 IF, 1:1000 WB), rabbit anti-Flag (Sigma-Aldrich, catalog# F7425, 1:100 IF, 1:1000 WB), mouse HECD1 anti-E-cadherin (gift from M. Takeichi and O. Abe, Riken Center for Developmental Biology, Kobe, Japan, 1:100 IF, 1:1000 WB), goat anti-E-cadherin (R&D Systems, catalog# AF748, 1:100 IF, 1:1000 WB), rabbit NW6 anti-DP (Green laboratory, 1:50 IF), chicken 1407 anti-Pg (Aves Laboratories, 1:2000 WB), rabbit anti-K5 and rabbit anti-K14 (gifts from J. Segre, National Human Genome Research Institute, 1:100 IF), rabbit anti-K17 (gift from P. Coulombe, Johns Hopkins University, 1:1000 IF, 1:5000 WB), mouse DM1α anti-α-tubulin (Sigma-Aldrich, catalog# T6199, 1:100 IF), rabbit anti-GAPDH (Sigma-Aldrich, catalog# G9545, 1:3000 WB), rabbit H191 anti-cortactin (Santa Cruz, catalog# sc-11408, 1:100 IF, 1:3000 WB), mouse JL8 anti-GFP (Clontech, catalog# 632381, 1:2000 WB) rabbit anti-GFP (Clontech, catalog# 632592, 1:2000 WB), mouse M2 anti-Flag (Sigma-Aldrich, catalog# F1804, 1:100 IF, 1:1000 WB), mouse anti-Arp3

**Fig. 5** Position of Dsg1 at the cell–cell interface regulates adherens junction contractility. **a** Immunofluorescence staining of vinculin and Flag (insets) in undifferentiated NHEKs retrovirally transduced with GFP, Dsg1-FL-Flag, or Dsg1-909-Flag. **b** Quantification of vinculin intensity at the cell borders of undifferentiated NHEKs expressing GFP, Dsg1-FL-Flag (FL), or Dsg1-909-Flag (909) (three independent experiments, at least 40 borders from each experiment were analyzed per condition; $F = 40.497$, ***$p < 0.001$ one-way Anova with Tukey test). **c** Structured illumination microscopy (SIM) images of vinculin and Flag in undifferentiated NHEKs expressing GFP, Dsg1-FL-Flag, or Dsg1-909-Flag (Flag staining, insets). **d** Quantification of the total area of vinculin puncta at the cell–cell borders from SIM images (three independent experiments; $F = 16.081$, ***$p < 0.001$ one-way Anova with Tukey test). **e** An E-cadherin FRET tension sensor (E-cad-TSmod) was co-expressed in undifferentiated NHEKs with mCherry, Dsg1-FL-Flag, or Dsg1-909-Flag. Immunofluorescence analysis for Flag, fluorescent images of the YFP channel demonstrating E-cad-TSmod expression and corresponding maps of the FRET index are shown. **f** FRET index for E-cad-TSmod was measured at cell–cell contacts in undifferentiated NHEKs expressing mCherry, Dsg1-FL-Flag (FL), or Dsg1-909-Flag (909) (five independent experiments; $F = 19.358$, ***$p < 0.001$ one-way Anova with Tukey test). **g** Western blot of undifferentiated NHEKs retrovirally transduced with E-cad-TSmod together with mCherry or Dsg1-FL-Flag and treated with control or Tctex-1 siRNA. GAPDH is a loading control. **h** Quantification of the FRET index for E-cad-TSmod at cell–cell junctions in undifferentiated NHEKs expressing mCherry or Dsg1-FL with or without silencing of Tctex-1 (four independent experiments; ns: non-significant; ***$p < 0.001$ unpaired two-tail $t$ test). Scale bars = 10 µm. Error bars represent SEM

(Abcam, catalog# ab49671, 1:100 IF), mouse hVIN-1 anti-vinculin (Sigma-Aldrich, catalog# V9131, 1:100 IF), mouse anti-p63 (Santa Cruz, catalog# sc-8431, 1:100 IF), mouse anti-CD104 (β4 integrin) (BD Biosciences, catalog# 611232, 1:100 IF), mouse C4 anti-actin (EMD-Millipore, catalog# MAB1501, 1:2000 WB), rabbit anti-Ki67 (EMD-Millipore, catalog# AB9260, 1:100 IF), rabbit anti-Survivin (Cell Signaling, catalog# 2808, 1:1000 IF), mouse anti-γ-tubulin (Abcam, catalog# ab11316, 1:100 IF), rabbit anti-Caspase-3 (Cell Signaling Technologies, catalog# 9662, 1:500 WB), mouse anti-Bcl-2 (Santa Cruz, catalog# sc-509, 1:500 WB) mouse G3G4 anti-BrdU antibody (DSHB, catalog# G3G4, 1:25 IF), rat DECMA anti-Ecadherin (Abcam, catalog# ab11512, 1:100 IF), rabbit anti-Beta catenin (Sigma, catalog# C2206, 1:200 IF).

Secondary antibodies used for western blotting included goat anti-mouse, -rabbit, -chicken HRP (Kirkegaard Perry Labs, 1:10,000 WB) and bovine anti-goat HRP (Santa Cruz, 1:10,000 WB). Secondary antibodies used for immunofluorescence included goat anti-mouse, -rabbit and -chicken conjugated with Alexa Fluor 488, 568 or 647 nm (Thermo Fisher Scientific, 1:300 IF); donkey anti-goat, -mouse, -rabbit conjugated with Alexa Fluor 488, 568, or 647 nm (Thermo Fisher Scientific, 1:300 IF); goat anti-mouse IgG1 isotype conjugated with Alexa Fluor 488, 568 or 647 nm (Thermo Fisher Scientific, 1:300 IF) and goat anti-mouse IgG2a isotype conjugated with Alexa Fluor 488 or 568 (Thermo Fisher Scientific, 1:300 IF). For staining of F-actin, Alexa Fluor 488, 568 or 647 nm phalloidin (Thermo Fisher Scientific, 1:50 IF) was incubated with secondary antibodies and with 4′,6-Diamido-2-Phenylindole (DAPI, Sigma-Aldrich, 1:500 IF) to stain nucleus.

**Chemical reagents**. For some experiments, keratinocytes were treated with either dimethyl sulfoxide (DMSO) (Sigma-Aldrich) or 100 μM CK666 (Sigma-Aldrich) for 4 h to inhibit Arp2/3 complex activity. To visualize lipid rafts in keratinocytes, the Vybrant Lipid Raft Labeling Kit (Thermo Fisher Scientific) was used according to the manufacturer's instructions: in which live keratinocytes were labeled with the fluorescent cholera toxin subunit B (CT-B) (labels gangliosides at the plasma membrane) by incubating cells with the CT-B conjugate working solution for 10 min at 4 °C.

For BrdU pulse chase, cells were incubated with BrdU labeling solution for 30 min at 37 °C, followed by PBS washes and 1 h incubation at 37 °C in cell culture medium. After fixation in 4% formalin for 15 min, cells were incubated in HCl solution (24 ml concentrated HCl in 60 ml of $H_2O$) for 10 min followed by washing three times in 0.5× TBE. Cells were further stained with antibodies as described in Immunofluorescence analysis and image acquisition.

TUNEL staining was performed according to the manufacturer's protocol (TUNEL Kit, Roche). The negative control was cells incubated with Label Solution without terminal transferase. The positive control was cells incubated with micrococcal nuclease for 10 min at room temperature prior to labeling procedures.

**Yeast 2-hybrid screening and interaction assays**. Screening was performed using the CytoTrap system (Agilent Technologies). Briefly, Cdc25H yeast were co-transformed with the Sos-fused Dsg1 bait constructs and a HeLa cell cDNA library carried in pMyr[41]; the procedure was performed per manufacturer's instructions (Aglient Technologies). Growth of yeast colonies at 37 °C on galactose plates suggested that yeast had positive interactors and these positive hits were subjected to further analysis. To confirm the interactions between pairs of proteins, the pMyr HeLa library was replaced by pMyr constructs coding specific proteins.

**Protein expression and pulldown**. pGEX-GST and pGEX GST-Dsg1 (cyto) (Dsg1 insert encoding the entire cytoplasmic domain) were transformed into Rosetta competent bacterial cells (Novagen). Cells were grown at 37 °C until $OD_{600} \approx 0.9$ and cooled to 22 °C. 100–200 μM of IPTG was added and the culture grew for another 4 h at 22 °C. Cells were harvested in lysis buffer (50 mM Tris-HCl, pH7.5, 100 mM NaCl, 5 mM $MgCl_2$, 0.5% Triton, protease inhibitor cocktail [Roche]). Lysates were sonicated on ice (eight times of 15 s burst with 30 s interval) and then centrifuged (10,000 × g for 20 min. at 4 °C). The cleared cell lysates were incubated with glutathione sepharose 4B (GE Healthcare) for 2 h at 4 °C. Beads with pulled down complexes were washed with PBS (4 × 10 min.).

Keratinocytes were lysed in 1 mL ice-cold lysis buffer (1% Triton, 145 mM NaCl, 10 mM Tris-HCl, pH 7.4, 10% glycerol, 5 mM EDTA, 2 mM EGTA, 1 mM phenylmethylsulfonyl fluoride and protease inhibitor cocktail [Roche] ± 1% phosphatase inhibitor cocktail IV [EMD-Millipore]). Lysates were incubated with equal amounts of GST/Dsg1 (cyto)-GST fused proteins for 2 h at 4 °C. Beads were extensively washed (3 × 10 min) with lysis buffer. Bound proteins were eluted in SDS/Laemmli sample buffer and were separated by 10% or 12.5% SDS-PAGE gels and analyzed by immunoblotting.

**Immunoprecipitation and immunoblotting**. Whole cell lysates were made using Urea Sample Buffer. Immunoprecipitation of ectopic FLAG-tagged or GFP-tagged Dsg1 constructs from cell lysates was carried out with M2-agarose beads (Sigma-Aldrich) or GFP-Trap beads (Chromo Tek). Cells were lysed in 1 mL of ice-cold lysis buffer (1% Triton, 145 mM NaCl, 10 mM Tris-HCl, pH 7.4, 10% glycerol, 5 mM EDTA, 2 mM EGTA, 1 mM phenylmethylsulfonyl fluoride and protease inhibitor cocktail [Roche] ± 1% phosphatase inhibitor cocktail IV [EMD]) and

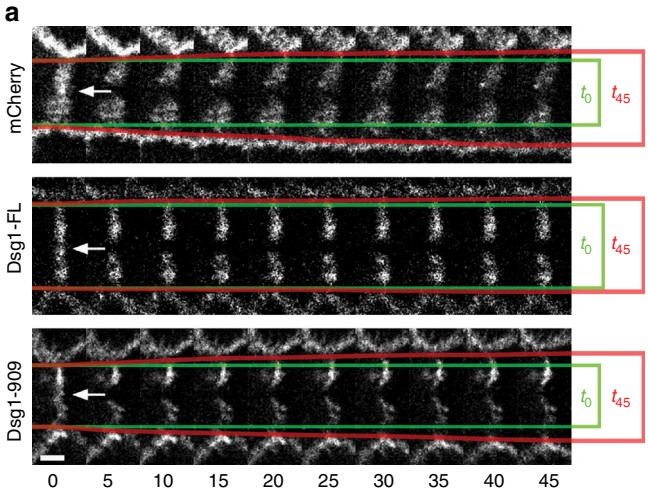

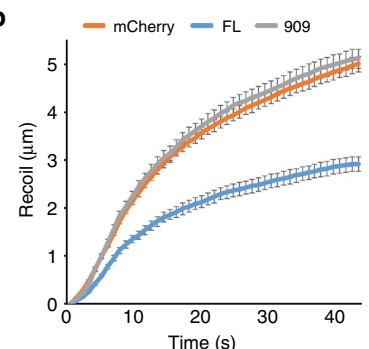

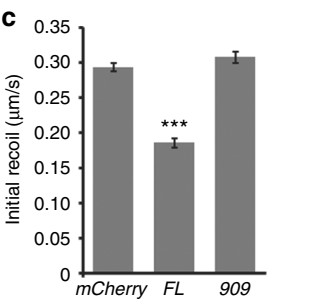

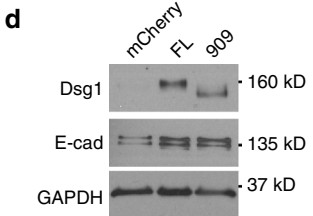

**Fig. 6** Position of Dsg1 at the cell–cell interface regulates cortical tension in keratinocytes. **a** Representative confocal images before and after ablation in NHEKs with mCherry, Dsg1-FL-Flag, or Dsg1-909-Flag. Scale bar = 5 μm. **b** Exponential curves of junctional recoil. **c** Quantification of tension (initial recoil) following ablation in NHEKs with mCherry, Dsg1-FL-Flag, or Dsg1-909-Flag (three independent experiments; $F = 110.5$, ***$p < 0.001$ one-way Anova with Tukey test). **d** Western blot of undifferentiated NHEKs retrovirally transduced with E-cad-YFP together with mCherry, Dsg1-FL-Flag, or Dsg1-909-Flag used for laser ablation. GAPDH is a loading control. Error bars represent SEM

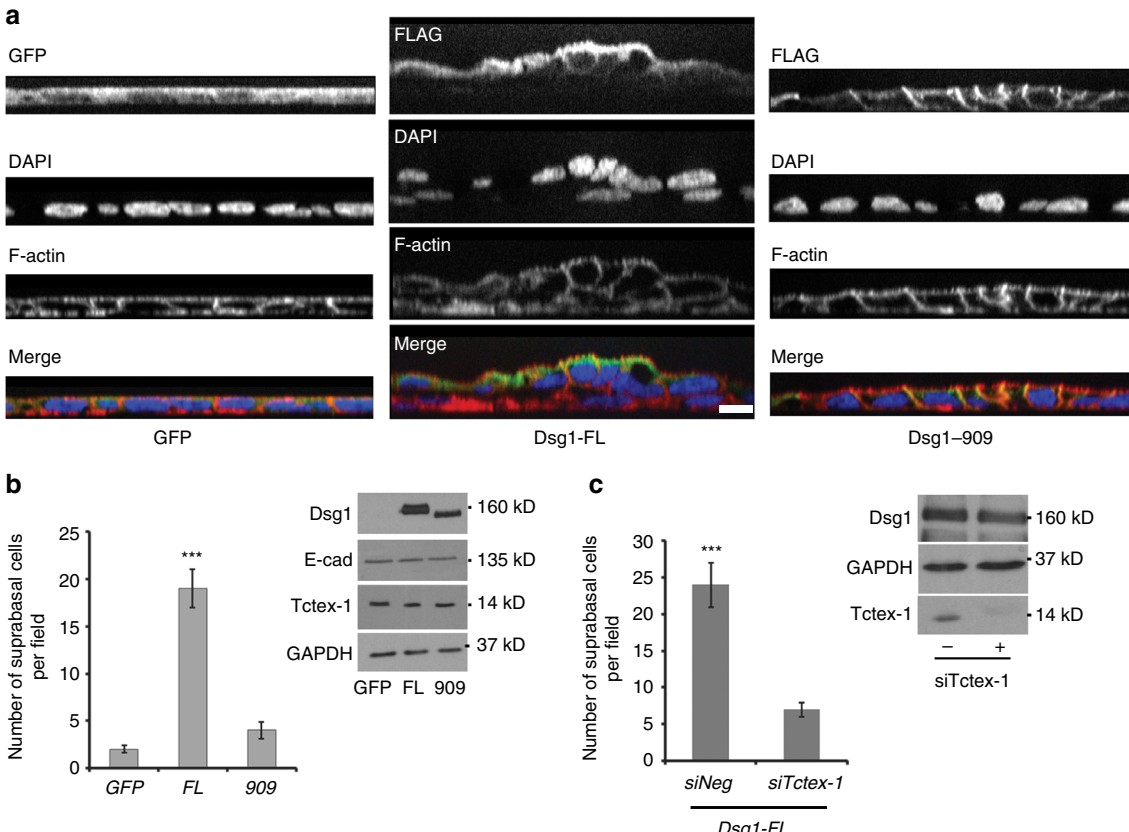

**Fig. 7** Tctex-1 dependent Dsg1 localization at the cell–cell interface promotes extrusion of MDCK cells from a polarized monolayer. **a** X–Z scanned images of MDCK polarized monolayers expressing GFP, Dsg1-FL-Flag, or Dsg1-909-Flag. Monolayers were stained for Flag (green), F-actin (red), and DAPI (marks nuclei). Scale bar = 20 μm. **b** Quantification of the amount of MDCK cells in a second layer on top of the monolayers expressing GFP, Dsg1-FL-Flag, or Dsg1-909-Flag (three independent experiments; $F = 59.503$, ***$p < 0.001$ one-way Anova with Tukey test). The levels of ectopic Dsg1 proteins expression and endogenous E-cadherin and Tctex-1 are shown on the corresponding western blot. GAPDH is loading control. **c** Quantification of the amount of MDCK cells remaining in a second layer on top of the monolayers expressing Dsg1-FL-Flag with or without Tctex-1 knockdown (three independent experiments; ***$p < 0.001$ unpaired two-tail $t$ test). The level of ectopic Dsg1-FL expression and Tctex-1 knockdown is shown on the corresponding western blot. GAPDH is loading control. Error bars represent SEM

centrifuged to remove debris. Cell lysates were then rotated with beads for 2 h at 4 °C, followed by washing and elution in SDS/Laemmli sample buffer. Cells expressing only GFP were used as a control. Proteins were separated by SDS-PAGE electrophoresis and transferred to nitrocellulose membranes. Immunoreactive proteins were visualized using enhanced chemiluminescence. Uncropped scans of the immunoblots are included in supplementary information (Supplementary Figures 11, 12, 13).

**Biotinylation assay.** Keratinocytes ectopically expressing Dsg1 constructs were washed two times in ice-cold PBS containing 1.2 mM Ca$^{2+}$ and incubated with 2 mg/mL EZ-Link Sulfo-NHS-SS-biotin (Pierce) for 30 min at 4 °C, protected from light. Excess biotin was removed with 100 mM Glycine in PBS by washing three times, 10 min each. Cells were washed a final time with PBS and lysed in RIPA buffer (10 mM Tris, pH 7.5, 140 mM NaCl, 1% Triton X-100, 0.1% SDS, 0.5% sodium deoxycholate, 5 mM EDTA and 2 mM EGTA, protease inhibitor [Roche]). Lysates were centrifuged at 14,000 rpm for 30 min, and cell supernatants were normalized for total protein. In total, 30 μl of each lysate was reserved for an input loading control. Biotinylated proteins were pulled down with 40 μl streptavidin beads (Pierce) at 4 °C overnight followed by several washes with RIPA buffer. Complexes were eluted using SDS/Laemmli sample buffer at 95 °C and analyzed by immunoblotting as described above.

**Subcellular fractionation.** Reagents for the subcellular fractionation analysis were purchased from Thermo Fisher Scientific and the assay was performed according to the manufacturer's protocol. As a final step, the leftover pellet was solubilized in Urea Sample Buffer to collect the desmosome-containing highly insoluble fraction. The amount of protein in all fractions was normalized and their distributions among the fractions were analyzed by immunoblotting as described above.

**Immunofluorescence analysis and image acquisition.** For immunofluorescence analysis, keratinocytes cultured on glass coverslips were fixed either in anhydrous

ice-cold methanol for 2 min at −20 °C or 4% formalin solution for 15 min at room temperature immediately followed by permeabilization with 0.1% Triton X-100 and 1 mg/ml bovine serum albumin (BSA) for 30 min. Fixed cells were incubated with primary antibodies for 30 min at 37 °C or overnight at 4 °C. After incubation with the primary antibodies, the coverslips were washed with PBS three times for 7 min followed by incubation with secondary antibodies for 30 min at 37 °C. After washing secondary antibodies, coverslips were mounted on glass slides with polyvinyl alcohol (Sigma-Aldrich). For analysis of organotypic epidermal raft 3D cultures, formalin-fixed, O.C.T-embedded tissue sections were processed by the Northwestern University Skin Disease Research Center. Slides were baked at 60 °C overnight and permeabilized with 0.5% Triton X-100 in PBS. Sections were blocked in blocking buffer (1% BSA/10% normal goat serum in PBS containing 0.1% Triton X-100) for 45 min at 37 °C. Normal goat serum was replaced with donkey serum or BSA when primary antibodies included goat IgG. Tissue sections were incubated with primary antibodies overnight at 4 °C in blocking buffer followed by three washes of PBS. Sections were incubated with secondary antibodies at 37 °C for 30 min. After three washes in PBS, coverslips were mounted using ProLong Gold Antifade Reagent (Thermo Fisher Scientific). Cells were visualized with epi-fluorescence microscopes (DMR; Leica or AxioVision Z1; Carl Zeiss) or at the Northwestern University Center for Advanced Microscopy utilizing a structured illumination super-resolution microscope (SIM, Nikon) or a Nikon A1R Confocal Laser microscope (Nikon). The DMR microscope images were generated using ×40 and ×63 objectives (PL Fluotar, NA1.0), a digital camera (Orca 100, model C4742–95; Hamamatsu) and MetaMorph 7.6 software (Molecular Devices, Sunnyvale). The AxioVision Z1 microscope images were generated using ×40 (Plan-Neofluar, NA0.5) and ×63 (Plan-Apochromat, NA1.4) objectives, a digital camera (Axio-Cam MRm; Carl Zeiss), an Apotome slide module (Carl Zeiss) and Axio-Vision software (Carl Zeiss). The SIM microscope was fitted with ×100 objective lens (NA 1.49; TiE SIM microscope) and iXon X3 897 camera (Andor Technology). The images were produced and analytically processed by Elements software (version 4, Nikon) to reconstruct the sub-resolution structure. Confocal images were acquired using a Nikon A1R Confocal Laser microscope equipped with GaAsP

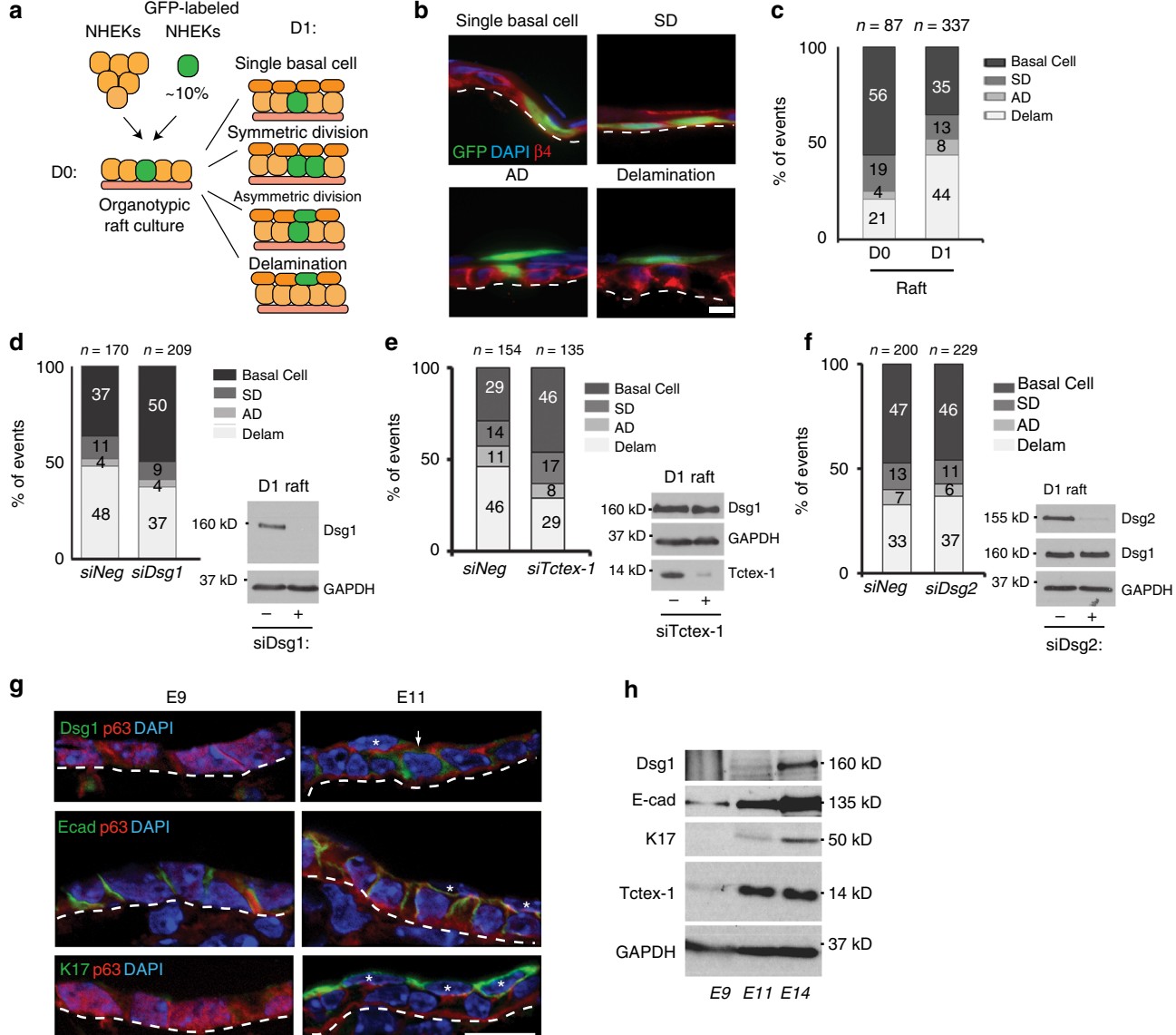

**Fig. 8** Tctex-1 dependent Dsg1 localization at the cell–cell interface promotes keratinocyte delamination. **a** Diagram of experimental design for analyzing early stratification events in 3D organotypic cultures (rafts) containing <10% GFP-positive cells. **b** The distribution of GFP-labeled cells used to track the events 24 h after initiation of stratification. Examples of observed events are shown. β4 integrin staining marks the basal layer, DAPI staining marks nuclei. **c–f** Bar graphs are population analyses showing the percentage of single basal cells (Basal Cell), symmetric division (SD), asymmetric division (AD) and delamination (Delam) observed in D0 (time of lifting to an air–liquid interface) and D1 rafts (one day after lifting to an air–liquid interface) (**c**), and D1 rafts treated with siNeg and siDsg1 (**d**), siTctex-1 (**e**), or siDsg2 (**f**) from at least five independent experiments for each condition. *n* represents the total number of events scored. Delamination events were significantly increased in D1 rafts compared to D0 (***$p < 0.001$, unpaired two-tailed *t* test). Knockdown of Dsg1 or Tctex-1 inhibit delamination (*$p < 0.05$ paired two-tailed *t* test), while Dsg2 knockdown did not have an effect. Immunoblots indicating levels of protein knockdown are shown to the right of each graph. **g** Representative fluorescence images of murine epidermis at days E9 and E11. Tissues were stained for Dsg1, E-cadherin, Keratin 17 (K17) (periderm marker), and p63 (ectoderm marker). DAPI staining marks nuclei. Dsg1-positive cells (arrowhead) were commonly seen associated with stratified periderm cells at E11 (asterisks). **h** Western blot analysis of Dsg1, E-cadherin, K17, and Tctex-1 expression levels in E9, E11, and E14 embryos. GAPDH serves as a loading control. Scale bars = 20 μm

detectors, a ×100 Plan-Apochromat objective lambda with an NA of 1.4, and run by NIS Elements software (Nikon). Images were further processed using Photoshop CS3 (Adobe) and compiled using Illustrator CS3 (Adobe).

**Proximity ligation assay**. Reagents for the PLA in situ were purchased from Sigma-Aldrich. PLA was performed according to the manufacturer's protocol (Duolink in Situ PLA, Olink Bioscience). In some cases chicken anti-plakoglobin followed by Alexa Fluor 488-conjugated goat anti-chicken secondary antibody (Thermo Fisher Scientific) was included after the PLA assay was completed to mark cell–cell borders. Fluorescent spots appearing at the site of Tctex-1/cadherin interactions and where applicable plakoglobin labeling were detected by wide-field microscopy (DMR; Leica) and compiled by ImageJ software (NIH).

**G-actin incorporation assay**. Keratinocytes expressing LZRS-GFP, LZRS-FLAG Dsg1 (FL), or LZRS-FLAG Dsg1 (909) constructs were incubated in M154 media with 1.2 mM Ca$^{2+}$ for 24 h. Cells were then equilibrated to room temperature (25 °C) for 10 min and subsequently permeabilized using 0.2 mg/mL Saponin in permeabilization buffer (138 mM KCl, 4 mM MgCl$_2$, 20 mM HEPES pH 7.4) for 1 min. Alexa 568 G-actin (0.45 μM, Thermo Fisher Scientific) in permeabilization buffer was added to the cells for 7 min at room temperature. Samples were fixed in 4% formalin solution with 0.2% Triton X-100 and Alexa 488 phalloidin (Thermo Fisher Scientific) for 1 h.

**FRET analysis**. Retrovirally transduced keratinocytes were plated on glass coverslips and incubated in M154 media with 1.2 mM Ca$^{+2}$ for 1, 6, or 24 h prior to

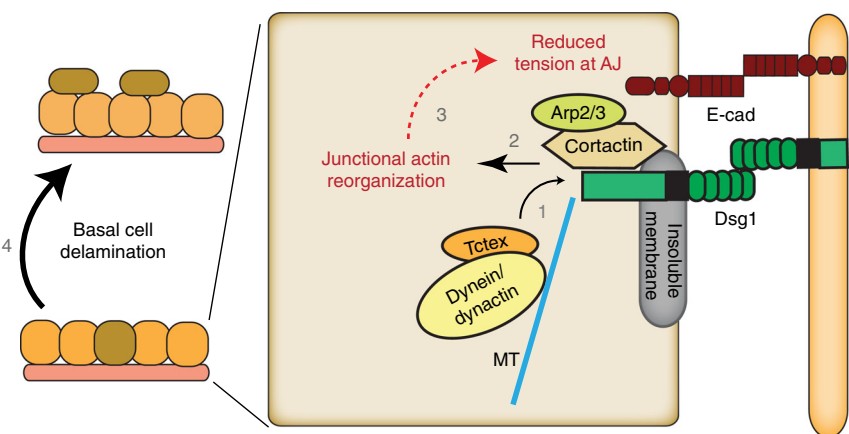

**Fig. 9** Diagram summarizing the findings of the present study. The sequence of events is required for initial steps of stratification. Newly synthesized Dsg1 is required for the Tctex-1–dynein complex to be localized at the cell–cell interface in the insoluble membrane compartment (1) where Dsg1 recruits cortactin–Arp2/3 actin nucleation complexes to promote actin polymerization/reorganization at the cell junctional area (2). Actin reorganization at the desmosomal junctions reduces tension at adherens junctions (AJ) (3) to promote basal cell delamination during the initial steps of stratification (4)

fixation and staining. Spinning disk confocal microscopy was performed on an Andor XDI Revolution microscope with a ×60 oil immersion CFI ×60 APO TIRF objective (NA 1.49) at the Northwestern University Center for Advanced Microscopy. MetaMorph software (Version 7.8.7, Molecular Devices, Sunnyvale CA) was used to obtain mTFP, eYFP, and RawFRET images with laser excitation at 445 nm for mTFP and RawFRET and 515 nm for eYFP. For mTFP a Semrock Brightline 480/40 filter was used and for eYFP and RawFRET a Semrock Brightline 534/30 filter was used for acquisition. Background subtracted, FRET/mTFP ratio images were generated using MetaMorph software. The equation used to calculate the FRET images was: $FRET = RawFRET - A \times mTFP - B \times eYFP$ where $A$ was the average calculated bleed through of E-cadherin-mTFP excited at 445 nm and imaged in the RawFRET channel, while $B$ was the average calculated excitation of E-cadherin-TSMod-eYFP at 445 nm and imaged in the RawFRET channel[47]. The average FRET/mTFP ratio at cell–cell contacts was quantified and normalized to the unit area. FRET/mTFP images shown were processed with a 3 × 3 median filter using ImageJ software to remove noise.

**Two-photon laser ablation**. NHEKs coexpressing either mCherry, mCherry-Dsg1, or Dsg1-909 and E-cadherin-eYFP were plated onto 60 mm glass bottom culture dishes (Willco Wells). Culture medium was changed to 1.2 mM Ca²⁺ for 24 h prior to imaging. Two-photon laser ablation was used to assess intercellular tension. Briefly, ablation was performed on a Nikon A1R-MP+ multiphoton microscope running Elements version 4.50 and equipped with an Apo LWD 25× 1.10W objective. Cells were maintained at 37 °C and 5% CO₂. Images of E-cadherin-eYFP were obtained at a rate of 1 frame per second for 2 s before and 45 s after ablation using 4% laser power at 920 nm. Ablation was performed using 40% laser power at a scan speed of 512. The distance between the cell–cell vertices over time was measured using the Manual Tracking plugin in ImageJ. Distance curves were then generated using Excel software. Initial recoil measurements were calculated using Prism software as previously described[48].

**Traction force measurements**. NHEKs were plated on collagen-coated poly-acrylamide gels with a shear modulus of 4.1 kPa, to analyze traction forces. Briefly, acrylamide/bis-acrylamide mixture formulated to produce the appropriate shear modulus and spiked with 40 nm fluorescent beads to track gel displacement was placed on a glass slide. Sulfo-SANPAH (Thermo Fisher) was utilized to conjugate collagen to gels. Cells were plated sparsely on the gels and allowed to grow for 2 days followed by a switch to 1.2 mM Ca²⁺ for 24 h. Images of live cell colonies and of the fluorescent beads were acquired. Then, cells were removed via treatment with 0.5% sodium dodecyl sulfate (SDS) and a second image of the beads was acquired to serve as a reference. After correcting for stage drift, gel deformation by cells was determined by comparing bead positions before and after SDS treatment utilizing MATLAB particle imaging velocimetry software (http://www.oceanwave.jp/softwares/mpiv/) which yielded a displacement field with a grid spacing of 1.08 μm. The Kriging interpolation method was utilized to filter and interpolate displacement vectors and traction stresses were reconstructed from the displacement field via Fourier Transform Traction Cytometry using zeroth order regularization[49,50]. The strain energy necessary to account for observed deformations was calculated for each colony and normalized to colony area to account for the effect of cell spreading on strain energy[51].

**Time-lapse imaging**. MDCK cells were seeded to confluency onto 24 mm Transwell inserts (0.4 μm pore, Costar) and grown 24 h before time-lapse imaging. The recording was done on a Nikon Biostation microscope that provides

environmental controls of temperature, humidity and gas concentration. Images were acquired at 5 min intervals for >12 h with a ×40 objective.

**Dispase mechanical dissociation assay**. Confluent cell cultures were rinsed with PBS and incubated with 2.4 U/ml dispase (Roche) for 30 min at 37 °C. Released monolayers were rotated on an orbital shaker (150 rpm) for 20 min prior to imaging. Fragments were visualized with a dissecting microscope (Leica MZ6) and the final images were obtained using a digital camera (Orca 100, model C4742-95; Hamamatsu). The numbers of fragments were counted by ImageJ (NIH) cell counter function and processed in Adobe Photoshop CS3.

**Animals and immunohistochemistry of embryos**. C57BL/6 wild-type mice were housed in accredited animal facilities and the use and care of mice were in accordance with the Northwestern University's Institutional Care and Use Committee. Embryos were collected from timed pregnant females and embryos were staged based on morphological characteristics and then processed for histological analysis. For immunohistochemistry, embryos were formalin-fixed for 4 h and followed by embedding in O.C.T. Compound. Tissue sections were processed by the Northwestern University Skin Disease Research Core. Slides were baked overnight at 60 °C and then were permeabilized using 0.5% Triton X-100 in PBS. Sections were blocked in blocking buffer (0.1% Triton X-100 and 1% BSA/10% normal goat serum in PBS) for 45 min at 37 °C. Goat serum was replaced with donkey serum or BSA when primary antibodies included goat IgG. Tissue sections were incubated with primary antibodies overnight at 4 °C in blocking buffer followed by washing in PBS. Secondary antibody incubation was done at 37 °C for 30 min. After three washes in PBS, coverslips were mounted using Prolong Gold Antifade Reagent (Thermo Fisher Scientific).

**Measurements, quantifications, graphing and statistics**. Cell area was determined by using ImageJ software (NIH). A minimum of 45 cells from three independent experiments (biological repeats) were measured. Fluorescence intensity of cell borders was determined by either measuring average intensity per area utilizing MetaMorph software (Version 7.8.7, Molecular Devices, Sunnyvale CA) or using the line scan function of ImageJ. For the line scan function, three independent areas with the same length and width were drawn at random positions for each border and average pixel intensity was collected. A minimum of 125 contacts from three independent experiments (biological repeats) were measured using Meta-Morph or ImageJ software. All images were captured using the exact same parameters. Pearson's correlation coefficients were calculated using NIS Elements software (Nikon).

For biochemical assays, at least three independent experiments were carried out and the numerical values for each band on the immunoblots were obtained using ImageJ software.

For all other assays, at least three independent experiments were carried out unless otherwise stated in the legends. Statistical analysis has been done using Excel, PRISM or MaxStat Lite software. For all experiments, data are means and error bars represent standard error of the mean (SEM); statistical analyses were performed using an unpaired or paired two-tailed *t* test or one-way Anova correction for multiple comparisons as detailed in the figure legends. *P* values less than 0.05 were considered statistically significant.

**Data availability**. The authors declare that all data supporting the findings of this study are available within the paper and its Supplementary Information files. In addition, all relevant data are available from the authors per request.

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

## Acknowledgements

We thank Carien Niessen and Matthias Ruebsam for helpful discussions and ideas. Gifts were provided by Pierre A. Coulombe (K17 antibody), Julie Segre (K5 and K14 antibody), Kevin Pfister (Tctex-1 antibody), and Alexander Dunn (Ecad-TSMod constructs). We are grateful to the Northwestern University Skin Disease Research Center (supported by the NIH/NIAMS; 5P30AR057216-08) for help with histological analysis of organotypic raft cultures. Any opinions, findings, and conclusions expressed in this material are those of the author(s) and do not necessarily reflect the views of the Northwestern University Skin Disease Research Center or NIH/NIAMS. We also thank the Northwestern University Center for Advanced Microscopy (generously supported by NCI CCSG P30 CA060553 awarded to the Robert H. Lurie Comprehensive Cancer Center) for assistance with imaging work. Structured illumination microscopy was performed on a Nikon N-SIM system, purchased through the support of NIH 1S10OD016342-01; FRET

analysis was performed on an Andor XDI Revolution microscope, purchased through the support of NCRR S10 RR031680-01; laser ablation experiments were performed on the A1R-MP+ multiphoton microscope, purchased through an S10 shared instrumentation grant awarded to Teng-Leong Chew (1 S10 OD010398-01); time-lapse imaging of MDCK cell extrusion was performed on Nikon Biostation system, acquired through the generous support from Northwestern University Office for Research and Skin Disease Research Center. This work was supported by NIH grants R01 AR041836, R37 AR043380 with partial support from R01 CA122151, the J.L. Mayberry Endowment to K. Green and the Chicago Biomedical Consortium with support from the Searle Funds at The Chicago Community Trust. O.N. was supported by a 2014 Dermatology Foundation Research grant. J.A.B. was supported by a Training Grant, T32 AR060710.

## Author contributions

O.N. designed study, conducted experiments, and analyzed data under the supervision of K.J.G. R.M.H. performed yeast work and generated the constructs. R.M.H. and M.L.G. performed traction force microscopy and analysis. J.A.B. performed G-actin incorporation experiments and laser ablation experiments and assisted with FRET experiments. J.L. K. provided 3D organotypic cultures and assisted with immunoblotting data acquisition. G.N.F. and L.M.G. performed animal work and provided embryos. O.N. and K.J.G. wrote the manuscript. All authors critically read and contributed to the manuscript.

## Additional information

**Competing interests:** The authors declare no competing interests.

