## [Peer Review File · Nature Communications]

Reviewers' comments:

Reviewer #1 (Remarks to the Author):

This manuscript reports the interaction of the desmosomal cadherin, desmoglein 1, with the dynein light chain Tctex-1. Multiple lines of evidence support this novel interaction which the authors further show is required for the proper organization of desmosomes. The underlying mechanism remains obscure, though Dsg1 and cortactin solubility were perturbed by loss of the Dsg1/Tctex-1 interaction. The rest of the manuscript presents evidence that Dsg1 recruits cortactin/Arp2/3 complex to the cell cortex where they promote F-actin assembly that results in decreased tension at adherens junctions. These findings are very interesting and provide the first demonstration of a mechanism for how desmosomes may regulate actin nucleation at the cortex (in a manner similar to adherens junctions). Finally, the authors propose that this mechanism is used to drive stratification of the epidermis during development. This is the most interesting part of the paper, however, it is also the least convincing.

At present there are a number of major concerns, and additional minor concerns, that prevent me from recommending this manuscript for publication.

Major Concerns

Mechanism of stratification.

The idea that Dsg1 promotes stratification is very intriguing, however, the data aren't compelling. There are a number of reasons for this. First, the discussion on how stratification occurs is both incomplete and misleading. The authors cite papers suggesting that delamination drives stratification, but did not discuss alternatives for which there is much more data, such as spindle orientation. There is significant controversy in the field over whether the delamination data is correct, especially as live imaging has not detected these events (at e14.5). Perhaps even more important is that the *in vivo* data on expression of Dsg1 in the mouse embryo looks at very early time points (e11.5). At this time periderm is being generated, not differentiated epidermis. The papers discussed to support the idea of delamination were all at the later stage and little is known about mechanisms promoting periderm formation. This could be very interesting and suggest that delamination drives periderm formation, but much more work would be required to validate that. Second, the assay system used to test a role for Dsg1 in delamination is flawed for two reasons. It does not begin with the same cell type (postnatal instead of embryonic) and thus it is difficult to believe that these cells would recapitulate the behavior of the embryonic cell types, especially as they do not give rise to periderm, but to differentiated epidermis. In addition, the assay has no "starting point" for comparison. We therefore cannot conclude that the suprabasal cells were generated by delamination rather than having never been attached. We also do not know how stratification occurs in these cultures (delamination versus spindle orientation) or whether Dsg1 is expressed in a subset of basal cells under these conditions.

Finally, the authors discuss how this may relate to extrusion-type delaminations in simple epithelia. But in those cases there are apical zonula adherens (with high tension) which may be quite different from epidermis where we have no data that forces are unequally distributed along the apical-basal axis. Would the same preferred type of apical extrusion occur under these conditions?

Role of Tctex-1. There is no mechanistic insight into how Tctex-1 might function at desmosomes. Two major questions related to this are, 1. Are lipid rafts affected by loss of Tctex-1 (as it appears in Figure 2h); and 2. Is Tctex-1 also at adherens junctions (the co-localization with Dsg1 does not seem very strong in Figure 1c).

Tension at adherens junctions. The data on changes in adherens junction tension also requires strengthening. Two lines of evidence currently support this idea – vinculin levels at junctions and

FRET tension sensor data. The vinculin data is clear, however, vinculin can also localize to junctions in tension-independent manners in some cell types and thus it is not conclusive. The FRET data is a much cleaner experiment, however, the data is not fully convincing. Can the authors please explain why cytoplasmic signal shows a decrease in tension (increase in FRET) in the Dsg1-FL expressing cells? The effects should only occur at the junctions, but there is a clear effect in the cytoplasm as well. The FRET image panel for Figure 5e also appears to be saturated (a lot of white) making it difficult to compare. Is the cytoplasm to junctional FRET ratio different between the samples?

Also – Dsg1 is also present in granular cells of the epidermis where tight junctions form and tension is thought to be high. The authors should discuss this point as well.

Minor Points

The authors use permeabilized cells to determine where actin assembly occurs. They refer to this as detecting barbed ends when it should reflect a combination of free barbed ends but also de novo nucleation.

How desmosome-promoted F-actin assembly results in a decrease in adherens junction tension remains vague.

Define domains of Dsg1 in Figure 1a.

Include co-localization of Tctex/cortactin with adherens junction markers. Perhaps desmosomes are required for cortical recruitment of Tctex/cortactin that is associated with adherens junctions. Along these lines, it would be useful to show Tctex and cortactin localization in differentiated Dsg1 KD cells to determine how significant the change in localization is (the current use of undifferentiated cells is useful, but there may be many more differences than Dsg1 levels).

Reviewer #2 (Remarks to the Author):

This paper suggests an intriguing role for the desmosomal cadherin desmoglein1 (Dsg1) in regulating actin polymerization and promotion of epidermal stratification. The interaction between Dsg1 and actin is believed to involve the dynein light chain Tctex-1 and cortactin/Arp2/3. These suggestions are based on some thorough experiments the data from which are generally well presented. The findings are novel and could be of wide interest. However, the results raise a number of questions which require consideration.

The resolution in Fig 1 C is rather poor, but staining for Dsg1 at the cell periphery appears punctate, as would be expected for desmosomes, while staining for Tctex-1 is linear. Is that because the Tctex picture is overexposed or because this protein actually has a linear distribution? If the latter, perhaps Tctex-1 is not so closely associated with Dsg1 in the cell as the binding studies might suggest. This might be resolved with better pictures, otherwise an explanation is required. Also these pictures were obtained from cells that had been allowed to differentiate, then treated with low Ca²⁺ medium overnight, then switched into medium with 1.2mM Ca²⁺ for 6 hours. All this is very artificial. What is the distribution of Dsg1 and Tctex-1 in epidermis? This should be easy to check and is highly relevant to the question of epidermal stratification? Fig 1D shows significantly more Tctex-1 in cells expressing Dsg1, but there is still some in cells lacking Dsg1, and in Fig1C this has a linear distribution at the cell periphery. Is that because there is some Dsg1 in cells that are claimed to lack it or because Tctex-1 has another binding partner at the periphery? If so, what is the significance of this? Strictly speaking, the various Dsg1 binding data in Fig1e-g should have Dsg2 controls.

Fig2b shows that knockdown of Tctex-1 causes a change in distribution of Dsg1. Is that just Dsg1 or is there a general disorganization of intercellular junctions? This should be easy to check and it

is relevant to the following results about protein solubility, which are confusing. Desmosomes are generally regarded as highly insoluble. Structure illumination microscopy is used to show that both Dsg1-FL and Dsg1-909, a mutant lacking the Tctex-1 binding region, colocalize with DP and this is stated to be consistent with their ability to incorporate into desmosomes. But Dsg1-909 is also shown to be predominantly in the soluble fraction. So does it incorporate into desmosomes or doesn't it? More important, it is shown that Tctex-1 knockdown results in increased solubility of Dsg1. So does the knockdown result in a general increase in desmosome solubility and thus potentially weakening of desmosomal adhesion? This could be quite important in relation of epidermal stratification.

The reorganization of perijunctional actin looks convincing. The data on peripheral tension from use of the E-cadherin tension sensor (E-cad-TS) are noted. In our experience the use of E-cad-TS is notoriously unreliable and difficult to interpret. Thus the results described might be the result of redistribution of E-cadherin rather than a change in tension. Confirmation of altered tension by and independent, more direct technique is necessary. This should be done by either cell stretching or traction force microscopy.

Although it is not quantified, the knockdown of Dsg1 in the stratification experiments shown in Fig 6 looks extremely efficient – there is none in the knockdown - yet the accompanying change in stratification is relatively small. How can this be if we are really dealing with the mechanism that regulates delamination?

Minor points:

The plural 'media' is used almost throughout when the intention seems to be the singular 'medium'.

Line 114: Since the authors have not actually defined a Tctex-1 binding site, would 'binding region' be a better term here?

Line 159: Surely 'different.....from' rather than 'differentthan'.

The pictures of mouse embryos in Fig 6 are superfluous.

RESPONSE TO REFEREES' COMMENTS

General introduction for all the referees. Specific responses to the verbatim critiques follow, and at the end is a comprehensive list of changes in the revised manuscript.

In this manuscript, we show that an actin remodeling complex associated with the evolutionarily recent cadherin, desmoglein 1 is necessary to remodel the cortical actin cytoskeleton, alter cortical tension and promote stratification of epithelial cells in contact. These data provide a novel perspective on how complex tissues emerged during evolution, and this previously unappreciated function for Dsg1 is likely to be of fundamental importance in tissue morphogenesis. In response to questions and issues raised by the two referees, we included new experiments in three major areas that together significantly strengthen the case for a specific role of desmoglein 1 in cortical tension and stratification. These include: a) additional experimental support to show that desmoglein 1 is both necessary and sufficient for epithelial stratification and promotes stratification in keratinocytes by delamination rather than asymmetric cell division, b) direct demonstration that desmoglein 1 mediates changes in junctional tension and c) elaboration Tctex-1's physical and functional relationship to desmosomes and desmoglein 1. We addressed these issues in the revised manuscript resulting in major revisions/additions to the figures, including the addition of one main figure, three new supplemental figures, and five new movies.

In the sections that follow, referees verbatim comments are included in bolded font, followed by our responses. In addition, a comprehensive outline of changes is provided at the end of the response to reviewers. Textual changes in the manuscript are denoted by yellow highlighting.

Reviewer #1 (Remarks to the Author):

This manuscript reports the interaction of the desmosomal cadherin, desmoglein 1, with the dynein light chain Tctex-1. Multiple lines of evidence support this novel interaction which the authors further show is required for the proper organization of desmosomes. The underlying mechanism remains obscure, though Dsg1 and cortactin solubility were perturbed by loss of the Dsg1/Tctex-1 interaction. The rest of the manuscript presents evidence that Dsg1 recruits cortactin/Arp2/3 complex to the cell cortex where they promote F-actin assembly that results in decreased tension at adherens junctions. These findings are very interesting and provide the first demonstration of a mechanism for how desmosomes may regulate actin nucleation at the cortex (in a manner similar to adherens junctions). Finally, the authors propose that this mechanism is used to drive stratification of the epidermis during development. This is the most interesting part of the paper, however, it is also the least convincing. At present there are a number of major concerns, and additional minor concerns, that prevent me from recommending this manuscript for publication.

Comment: *Mechanism of stratification.* The idea that Dsg1 promotes stratification is very intriguing; however, the data aren't compelling. There are a number of reasons for this. First, the discussion on how stratification occurs is both incomplete and misleading. The authors cite papers suggesting that delamination drives stratification, but did not discuss alternatives for which there is much more data, such as spindle orientation. There is significant controversy in the field over whether the delamination data is correct, especially as live imaging has not detected these events (at e14.5). Perhaps even more important is that the in vivo data on expression of Dsg1 in the mouse embryo looks at very early time points (e11.5). At this time periderm is being generated, not differentiated epidermis. The papers discussed to support the idea of delamination were all at the later stage and little is known about mechanisms promoting periderm formation. This could be very interesting and suggest that delamination drives periderm formation, but much more work would be required to validate that. Second, the assay system used to test a role for Dsg1 in delamination is flawed for two reasons. It does not begin with the same cell type (postnatal instead of embryonic) and thus it is difficult to believe that these cells would recapitulate the behavior of the embryonic cell types, especially as they do not give rise to periderm, but to differentiated epidermis. In addition, the assay has no "starting point" for comparison. We therefore cannot conclude that the suprabasal cells were generated by delamination rather than having never been attached. We also do not know how stratification occurs in these cultures (delamination verses spindle orientation) or whether Dsg1 is expressed in a subset of basal cells under these conditions.

Finally, the authors discuss how this may relate to extrusion-type delaminations in simple epithelia. But in those cases there are apical zonula adherens (with high tension) which may be quite different from epidermis where we have no data that forces are unequally distributed along the apical-basal axis. Would the same preferred type of apical extrusion occur under these conditions?

Response: We appreciate that the reviewer finds this first demonstration for how desmosomes may regulate actin nucleation at the cortex as a mechanism to drive stratification of the epidermis during development interesting. While they found the concept that desmoglein 1 (Dsg1) promotes stratification the most interesting, they also found it the least convincing. For this reason we have spent considerable effort developing both completely new strategies, as well as adding new experimental arms to existing strategies to demonstrate that Dsg1 is both necessary and sufficient to promote epithelial stratification.

Dsg1 promotes stratification/extrusion in simple epithelial cells: A major addition to the paper comes from experiments in which ectopic Dsg1 proteins were introduced into simple polarized cells. These data address the final question in the paragraph above: *Would the same preferred type of apical extrusion occur in simple epithelium under these conditions?* But we discuss them first as they provide an important foundation for the rest of the discussion and compelling support for the idea that Dsg1 is sufficient to promote extrusion of cells from the monolayer (**Figure 7 and Supplementary Video 4 and 5**).

To address this question we utilized MDCK cells, which are simple epithelial cells that express endogenous Tctex-1 and cortactin but never express Dsg1 and do not stratify under normal conditions. MDCK cells were transfected with either Dsg1-FL or the Tctex binding-deficient mutant Dsg1-909, or GFP as a control. Our data show that Dsg1-FL expression leads to more extrusion events from the monolayer than Dsg1-909 or GFP. Furthermore, extruded Dsg1-expressing cells remain at the top of cell monolayer and form small focal areas resembling stratifying tissue. Moreover, knockdown of Tctex-1 in MDCK cells expressing Dsg1-FL decreased extrusion events to the level of Dsg1-909 expressing cells and the GFP control, suggesting that Dsg1-Tctex-1 interactions are required for promoting apical extrusion from the monolayer.

Cell extrusion in simple epithelia can be increased due to apoptotic stimuli within the monolayer to repair the barrier or when epithelial cells become too crowded due to an increase in proliferation (Gudipaty and Rosenblatt, *Semin Cell Division Biology*, 2016 S1084-9521(16)30136-7). To rule out the possibility that alterations in cell proliferation contributes to the observed extrusion events in Dsg1-FL expressing MDCK cells, we performed a BrdU pulse labeling experiment in the Dsg1-FL, Dsg1-909 or GFP positive monolayers, and quantified mitotic cells. Neither BrdU incorporation nor the number of Ki67-positive cells were significantly altered under these conditions. **We included these data in Supplemental Figure 8a, b.** Next, we analyzed the rate of apoptosis within the monolayer of GFP, Dsg1-FL or Dsg1-909 positive cells, utilizing the TUNEL assay and immunoblotting for Bcl-2 and Cleaved Caspase 3 in total cell lysates. We did not detect any signs of increased apoptosis in Dsg1-FL expressing cells compared to the GFP-control and Dsg1-909 expressing cells. **We added these data in Supplemental Figure 8c, d.**

In conclusion, these data provide a dramatic demonstration of the morphogenetic potential of Dsg1, *and provide proof-of-principle that by coupling to endogenous machinery already present in simple epithelia, the evolutionarily advanced cadherin Dsg1 is sufficient to promote delamination out of the monolayer. They provide us with an entirely new perspective how cells might form multi-layered tissues, by identifying a new cell biological “tool” that cells can use to escape/extrude from a cell sheet.*

Dsg1 promotes delamination but doesn't affect spindle orientation: The referee also raised a series of issues regarding delamination, whether Dsg1 is in fact promoting this process, the extent to which delamination is relevant at specific stages of embryogenesis, and the failure to adequately consider other mechanism such control of spindle orientation.

With respect to spindle orientation, we would like to thank the reviewer for the suggestion to study in more detail whether asymmetric cell division was affected when Dsg1-Tctex-1 interaction is abrogated. This is an important issue, as it has been shown that changes in the actin cytoskeleton and cell shape are associated with efficient spindle re-orientation in keratinocytes (Gillies and Cabernard, *Current Biology* 2011 21:R599-609; Mao et al., *Genes & Development* 2011 25:131-6). While we had included a partial analysis of this issue in the original version of the manuscript, we have expanded it in the revised manuscript, adding additional controls and methods of analysis. First, we assessed the number of symmetric and asymmetric

events in our 3D epidermal organotypic cultures in control, Dsg1, Tctex-1 or Dsg2 (used as a control) knockdown conditions. Silencing Dsg1, Tctex-1 or Dsg2 did not affect the number of symmetric or asymmetric divisions. **These data are shown in Figure 8 panel d-f (former Figure 6 panel e-g).** As an alternative approach, we stained for γ -tubulin as a marker of the centrosome to check for defects in cell polarity. The data show that knockdown of Dsg1 or Tctex-1 does not affect apical position of centrosomes relative to the nucleus in basal keratinocytes, suggesting that basal cell polarity is not affected under these conditions. We also assessed the position of the cleavage furrow marker survivin as an indicator of epidermal division angles. Based on the analysis of survivin angle, a large majority of cell divisions in control and knockdown D1 organotypic raft cultures occur with the spindle oriented parallel to the basement membrane, which supports quantifications shown in Figure 8 and correlates with previously observed data at E12.5-13.5 in mouse embryos (Williams et al., Nature Cell Biology 2014). **These new data are shown in Supplemental Figure 9a-d.** Altogether the above data provide strong support that a decrease in stratification that occurs in Dsg1 or Tctex-1 knockdown cultures does not depend on interference with cell polarization or spindle orientation.

The referee was also concerned that there was no “starting point” for comparison in this same keratinocyte delamination assay. We strongly agree and thank the referee for bringing this up. To address this concern, we added a Day 0 baseline experiment to calculate the number of single superficial cells present in the population at first plating, in order to be able to essentially subtract out that component. Specifically, we compared D0 and D1 (24h of stratification) rafts, assessing the percentage of single basal cells, symmetric divisions, asymmetric divisions and delaminations. We detected 21% of single GFP labeled cells sitting on the top of the monolayer at the “starting time point”. However this number significantly increased after 24 hours of stratification to the 44% while the number of single basal cells was significantly decreased from 56% to 35% suggesting that basal cells underwent delamination during the 24h of stratification. **These data are presented in Figure 8c.** Together with the significant decrease in delamination events observed at Day 1 under Dsg1 or Tctex-1 knockdown conditions, these data provide compelling support for the importance of Dsg1-Tctex-1 interactions in promoting efficient delamination in this model system. (The referee may also want to refer to response to referee #2 regarding this issue).

The reviewer also raised the issues that a) the cells we used in the keratinocyte delamination assay are postnatal, not embryonic, and b) the contribution of delamination to epidermal development are controversial and delamination events have not been detected by live cell imaging at stage e14.5 (which, to our knowledge, has not been published). While we do show for the first time that Dsg1 expression is temporally coordinated with the first stratification event (i.e. periderm formation) and is thus consistent with a role in this process, we recognize that our data does not establish the extent to which the observed Dsg1-driven morphogenesis contributes to specific stages of epidermal development or to adult homeostasis. An evaluation of when and how during development and/or the adult such a mechanism is employed would require a more thorough analysis than is possible in this manuscript. *However, what our data do convincingly establish is that Dsg1 is necessary and sufficient to drive stratification: necessary for efficient delamination of keratinocytes at the onset of differentiation and sufficient for extrusion of simple epithelial cells that normally never express this cadherin, as long as the endogenous machinery necessary to couple it to the actin remodeling system is present. Identification of this previously unrecognized function for Dsg1 is an advance with broad implications for mechanisms of complex tissue morphogenesis.* Our data provide a valuable new perspective and a framework for future analysis of the different temporal and spatial contexts where stratification is occurring in vivo, whether it is during periderm formation or later in development or in the adult. We have revised our paper to be careful not to over-interpret our data and to be explicit about what we can conclude from the model systems used in this work.

Comment: Role of Tctex-1. There is no mechanistic insight into how Tctex-1 might function at desmosomes. Two major questions related to this are, 1. Are lipid rafts affected by loss of Tctex-1 (as it appears in Figure 2h); and 2. Is Tctex-1 also at adherens junctions (the co-localization with Dsg1 does not seem very strong in Figure 1c).

Response: In response to the general question about mechanism by which Tctex-1 assists Dsg1 in mediating its actin remodeling function (response to specific questions follow), our data support the following: 1) Tctex-1 light chain is not required for Dsg1's plasma membrane delivery, but based on immunofluorescence and biochemical fractionation analysis it is required for properly positioning Dsg1 on the membrane in a domain distinct from E-cadherin (**Figure 3e (former Figure 2f)**); 2) Dsg1-Tctex-1 interactions are required to efficiently

co-localize with a fluorescently labeled ganglioside lipids, a marker for lipid rafts with which desmosomes have previously been reported to have an association (Nava et al., Mol Biol Cell 2007, 11: 4565-4578; Resnik et al., J Bio Chem, 2011 286:1499-1507) (**Fig 3g**; see more on lipid rafts below) and 3) Tctex-1-dependent Dsg1 positioning at the plasma membrane and within the correct biochemical fraction requires Tctex-1 association with the dynein motor based on analysis of Tctex-1 mutants that have been shown to uncouple from dynein (**Figure 3i-h**; **Supplementary Figure 3b-c**, **Supplementary Figure 6b**). The fact that the dynein motor protein is known to be involved in lipid raft-enriched vesicle delivery to the plasma membrane (Hanzal-Bayer and Hancock, FEBES letters 2007, 581 2098-2104), is also consistent with the possibility that the Tctex-1-dynein complex targets Dsg1 to a lipid raft-like membrane compartment. But how the Tctex-1-dynein complex mediates Dsg1 localization, for instance through localized recycling or clustering events, is an important question that deserves a thorough analysis and multiple lines of experimentation beyond what we can reasonably include in this paper.

Importantly, we can also say that the mis-positioning of Dsg1 due to loss of Tctex-1 binding does not affect Dsg1's association with desmoplakin or impair the strength of intercellular adhesion within these cultures (**Supplementary Figure 2b-d**). Thus, while Dsg1-Tctex-1 interactions are required for dynein-dependent insoluble membrane compartmentalization of Dsg1 at the plasma membrane, interfering with this interaction does not impair Dsg1's association with the desmosomal plaque or affect desmosomal adhesive function or epithelial sheet integrity on a global level.

Response to specific questions:

Is Tctex at adherens junctions: To assess the extent to which Tctex-1 and E-cadherin (as a marker of adherens junctions) versus Tctex-1 and Dsg1 interact, we: 1) compared co-localization between Dsg1-Tctex-1

Figure 1 for referee: E-cadherin and Dsg1 immunoprecipitation from differentiated NHEKs probed for b-catenin (positive control for E-cadherin) and Tctex-1.

and E-cadherin-Tctex-1 by utilizing the Pearson correlation coefficient (shown in **Figure 2a and b**), 2) used proximity ligation assay (shown in **Figure 2c and d**), and 3) determined whether Tctex-1 associates with Dsg1 or E-cadherin in differentiated keratinocytes by co-immunoprecipitation (shown here for the referee's information; **Figure 1 for referees**). Altogether, the data support the conclusion that Tctex-1 specifically associates with Dsg1 and not with E-cadherin in human keratinocytes. Additionally, we addressed whether Tctex-1 knockdown affects E-cadherin distribution at cell-cell interfaces to the same extent as it does for Dsg1 distribution. E-cadherin distribution was not changed in Tctex-1 deficient cells compared to the control, while Dsg1 within cell contacts was more broadly distributed and disorganized under knockdown conditions compared to control (**data shown in Figure 3a and b**). We also refer the referee to the response to referee #2, for additional controls related to the specificity of the Dsg1-Tctex interaction.

Are lipid rafts affected by loss of Tctex: To address this question, lipid rafts were isolated from differentiated primary human keratinocytes with or without Tctex-1 knockdown utilizing a 5-40% sucrose gradient assay. Immunoblotting showed that the distribution of lipid raft markers flotillin-1 and caveolin-1 were not significantly perturbed in Tctex-1 knockdown cells, supporting the conclusion that the rafts are not perturbed using this assay. Because of the limited amount of Dsg1 that is solubilized in this procedure (Dsg1 is notoriously insoluble) and dilution across fractions, we did not have enough solubilized Dsg1 proteins (endogenous or ectopic, not shown) to convincingly assess their distribution relative to rafts utilizing this technique (**Figure 2 for referees**), so we have limited the data in the paper to use of the fluorescence lipid raft marker (Figure 3g) and cell fractionation data (Figure 3e), and include the sucrose gradient data for the referees' information. Overall, the data support the idea that Tctex-1 localizes Dsg1 to membranes that share properties with lipid rafts, thus appropriately positioning the cortactin/Arp2/3, but does not disrupt rafts.

Figure 2 for referees. Isolation of membrane rafts utilizing 5-40% sucrose gradient. Top panel: differentiated NHEKs with or without Tctex-1 knockdown were lysed with 1% Triton X-100 and lipid rafts were isolated by ultracentrifugation. Flotillin-1 and Caveolin-1 are markers of lipid rafts. DRM stands for detergent resistant membranes. Protein in each fraction was normalized to total protein and graphed below. The overlapping profiles indicate that the raft markers are not altered by Tctex-1 knockdown.

that E-cadherin is under constitutive tension at the plasma membrane independent of localization to cell-cell contacts. This tension is generated by E-cadherin's connection to actin cytoskeleton. Therefore, any changes in actin cytoskeleton structure transmit to the changes in plasma membrane bound E-cadherin. In our work, we showed that Dsg1-FL expression induces cortical actin reorganization in cells, which is associated with changes in E-cadherin FRET signal at cell-cell contacts. To address the reviewer's question about "cytoplasmic signal" and confirm observations that were made by Borghi, et al, we quantified FRET index for E-cadherin-TS outside of the cell contacts and found that in Dsg1-FL positive cells FRET index increases up to 40+/-0.4 compared to mCherry controls and Dsg1-909 expressing cells which are 32+/-0.3 and 37+/-0.4 under those conditions, suggesting that Dsg1-FL expression has a global effect on the plasma membrane tension distribution rather than being restricted to the cell junctional area. The statement was added to the main text.

In addition, in response to referee #2's suggestions we added two new experimental approaches to provide more direct evidence for a Dsg1-mediated change in cortical tension when Dsg1-FL is expressed in keratinocytes: two-photon laser ablation and traction force microscopy (see also response to referee #2). Laser ablation at E-cadherin-labeled junctions was carried out to allow measurement of contractile properties of the cells when mCherry-control, Dsg1-FL or Dsg1-909 mutant were expressed. Measurements were done by recording instantaneous recoil of labeled membrane from the cut site (**Supplemental Videos 1-3**). The degree of recoil provides an indicator of the level of tension in the cells. The data show that cells with Dsg1-FL have significant slower recoil compared to mCherry-control or Dsg1-909 mutant expressing cells, signifying less tension. **We added these data to Figure 6i-k.** See response to referee #2 for details of the traction force microscopy experiment. These data complement the original FRET and vinculin-staining analysis, and altogether, an even more compelling argument for Dsg1-dependent decreases in cortical tension at the keratinocyte cell membrane.

Comment: Tension at adherens junctions. The data on changes in adherens junction tension also requires strengthening. Two lines of evidence currently support this idea – vinculin levels at junctions and FRET tension sensor data. The vinculin data is clear, however, vinculin can also localize to junctions in tension-independent manners in some cell types and thus it is not conclusive. The FRET data is a much cleaner experiment, however, the data is not fully convincing. Can the authors please explain why cytoplasmic signal shows a decrease in tension (increase in FRET) in the Dsg1-FL expressing cells? The effects should only occur at the junctions, but there is a clear effect in the cytoplasm as well. The FRET image panel for Figure 5e also appears to be saturated (a lot of white) making it difficult to compare. Is the cytoplasm to junctional FRET ratio different between the samples?

Response: We would like to thank the reviewer for noticing the change in the FRET signal for E-cadherin outside of the cell junctions. Indeed, it has been reported before that E-cadherin transduces mechanical forces through its cytoplasmic domain not only at cell-cell contacts but also at contact-free plasma membrane throughout the cell. Thus Borghi et al. (PNAS 2012, vol.109 12568-12573) compared FRET index for E-cad-TS at the cell-cell contacts and the contact-free plasma membrane area. They showed

Comment: Dsg1 is also present in granular cells of the epidermis where tight junctions form and tension is thought to be high. The authors should discuss this point as well.

Response: The reviewer makes a very important point. To address this issue, we have clarified in the text that the current study describes the molecular mechanism which plays a role only during initial steps of stratification when cells are leaving basal layer (line 375 and 430 in the main text). The facts that Tctex-1 is predominantly expressed and co-localized with Dsg1 in the basal and first suprabasal layers of epidermis (**panels f and g in Figure 2**) further support this hypothesis.

Minor Points

Comment: The authors use permeabilized cells to determine where actin assembly occurs. They refer to this as detecting barbed ends when it should reflect a combination of free barbed ends but also de novo nucleation.

Response: Thank you for pointing this out; we made the changes to the text.

Comment: How desmosome-promoted F-actin assembly results in a decrease in adherens junction tension remains vague.

Response: We have included more explicit discussion of our model in the Discussion (lines 422-432) of the main text. Our data suggest that Dsg1-Tctex-1 interactions promote remodeling of two actin populations in cells: 1) it causes the loss of orthogonally-oriented cell-contact-associated F-actin bundles, which leads to decrease in actomyosin contractility (please see the **new Supplementary Figure 5c, d**) and 2) recruits cortactin-Arp2/3 complex to the junctional area to polymerize actin at Dsg1-positive desmosomes. Changes in both actin populations at the cell-cell interface are likely to contribute to the observed decrease in tension at adherens junctions.

Comment: Define domains of Dsg1 in Figure 1a.

Response: Thank you for this suggestion, domains were defined in the figure legend.

Comment: Include co-localization of Tctex/cortactin with adherens junction markers. Perhaps desmosomes are required for cortical recruitment of Tctex/cortactin that is associated with adherens junctions. Along these lines, it would be useful to show Tctex and cortactin localization in differentiated Dsg1 KD cells to determine how significant the change in localization is (the current use of undifferentiated cells is useful, but there may be many more differences than Dsg1 levels).

Response: As described above, our data demonstrate that Tctex-1 specifically interacts with Dsg1 and does not co-localize or associate with adherens junction markers (**Figure 2a-d**).

We also added an E-cadherin immunoblot to **Figure 5a** to show that Dsg1 associates with cortactin without E-cadherin in the complex. Finally, as suggested by the referee, we performed immunofluorescence staining for cortactin localization in differentiated Dsg1 KD cells (please see the new **Supplementary Figure 6c**). Knockdown of Dsg1 abolished cortactin accumulation at the junctional area in differentiated NHEKs.

Reviewer #2 (Remarks to the Author):

This paper suggests an intriguing role for the desmosomal cadherin desmoglein1 (Dsg1) in regulating actin polymerization and promotion of epidermal stratification. The interaction between Dsg1 and actin is believed to involve the dynein light chain Tctex-1 and cortactin/Arp2/3. These suggestions are based on some thorough experiments the data from which are generally well presented. The findings are novel and could be of wide interest. However, the results raise a number of questions which require consideration.

Comment: The resolution in Fig 1 C is rather poor, but staining for Dsg1 at the cell periphery appears punctate, as would be expected for desmosomes, while staining for Tctex-1 is linear. Is that because the Tctex picture is overexposed or because this protein actually has a linear distribution? If the latter, perhaps Tctex-1 is not so closely associated with Dsg1 in the cell as the binding studies might suggest. This might be resolved with better pictures, otherwise an explanation is required. Also these pictures were obtained from cells that had been allowed to differentiate, then treated with low Ca²⁺ medium overnight, then switched into medium with 1.2mM Ca²⁺ for 6 hours. All this is very artificial. What is the distribution of Dsg1 and Tctex-1 in epidermis? This should be easy to check and is highly relevant to the question of epidermal stratification? Fig 1D shows significantly more Tctex-1 in cells expressing Dsg1, but there is still some in cells lacking Dsg1, and in Fig1C this has a linear distribution at the cell periphery. Is that because there is some Dsg1 in cells that are claimed to lack it or because Tctex-1 has another binding partner at the periphery? If so, what is the significance of this? Strictly speaking, the various Dsg1 binding data in Fig1e-g should have Dsg2 controls.

Response: The referee raises an important point. To better address the proximity and specificity of the Tctex-Dsg1 interaction we carried out a series of additional experiments and controls. First, we repeated immunostaining of Tctex-1 and assessed its co-localization with Dsg1 and Dsg2 at the cell junctional area utilizing Pearson correlation coefficient. Our data showed that Tctex-1 accumulates with Dsg1 but not with Dsg2 at the cell junctional area. **We included the data to the Figure 2a, b.** Next, we assessed interactions between desmosomal cadherins and Tctex-1 in endogenous complexes in situ using the proximity ligation assay. Utilizing this method, we provide data to demonstrate close proximity of endogenous complexes specifically containing Dsg1 and Tctex-1. **These data are shown in Figure 2c, d.** The referee is correct that the staining of Tctex-1 is more linear than that of Dsg1, and we full agree that it is possible that Tctex-1 can bind other proteins at the plasma membrane that have not been studied in this current work. But our data provide strong support that Tctex-1 interacts with Dsg1 but not with Dsg2 at the cell junctions in human keratinocytes and that interfering with Dsg1-Tctex-1 interactions is functionally significant.

The referee also makes a very good suggestion to look at the distribution of Dsg1 and Tctex-1 in human epidermis. Utilizing structured illumination microscopy we showed that Dsg1 and Tctex-1 co-localize at the cell-cell interface in the basal layer of the epidermis further supporting the idea that this functional complex can exist under physiological conditions. **These data are shown in Figure 2f, g.** It is important to note that Tctex-1 is concentrated in the basal layers where the initiation of stratification is occurring, supporting its functional importance in these early stages of differentiation.

Comment: Fig2b shows that knockdown of Tctex-1 causes a change in distribution of Dsg1. Is that just Dsg1 or is there a general disorganization of intercellular junctions?

Response: In response to this query we assessed Dsg2 and E-cadherin distribution at cell-cell interfaces in control and Tctex-1 knockdown cells relative to Dsg1 distribution. Both E-cadherin and Dsg2 accumulation and organization at the junctions were unaltered in Tctex-1-deficient cells suggesting that Tctex-1 knockdown affects specifically Dsg1 positive desmosome localization. We included this data to **Figure 3a** (former Figure 2 panel b). We also replaced the border intensity quantifications in **Figure 3b** (former Figure 2 panel c) with line scan analysis of borders to better represent the change in Dsg1 border organization in Tctex-1 knockdown cells.

Comment: Structure illumination microscopy is used to show that both Dsg1-FL and Dsg1-909, a mutant lacking the Tctex-1 binding region, colocalize with DP and this is stated to be consistent with their ability to incorporate into desmosomes. But Dsg1-909 is also shown to be predominantly in the soluble fraction. So does it incorporate into desmosomes or doesn't it? More important, it is shown that Tctex-1 knockdown results in increased solubility of Dsg1. So does the knockdown result in a general increase in desmosome solubility and thus potentially weakening of desmosomal adhesion?

Response: The referee makes an excellent point regarding the role Dsg1-Tctex-1 binding to Dsg1-positive desmosome assembly and desmosome adhesive function. To address the functional issue, we utilized a mechanical dissociation assay (dispass assay), in which a confluent monolayer of cells was enzymatically released from the cell culture dish and subjected to mechanical stress to generate fragments of the epithelial sheet. Expression of Dsg1-FL or Dsg1-909 mutant increased intercellular adhesive strength in non-

differentiated human keratinocytes compared to the GFP control, suggesting that both of them can form functional desmosomes. Moreover, silencing Tctex-1 in differentiated keratinocytes did not alter cell-cell adhesion, compared to the positive control, desmoplakin knockdown. We include these data in **Supplemental Figure 2d**. Together with structured illumination microscopy, which showed Dsg1-909 co-localization with DP, these data provide compelling support for the ability of Dsg1 to incorporate into desmosomes that retain their adhesive function when its interaction with Tctex-1 is abrogated.

Comment: The reorganization of perijunctional actin looks convincing. The data on peripheral tension from use of the E-cadherin tension sensor (E-cad-TS) are noted. In our experience the use of E-cad-TS is notoriously unreliable and difficult to interpret. Thus the results described might be the result of redistribution of E-cadherin rather than a change in tension. Confirmation of altered tension by and independent, more direct technique is necessary. This should be done by either cell stretching or traction force microscopy.

Response: The referee makes a great suggestion to incorporate a more direct test of altered tension. Toward this end, we utilized two-photon laser ablation as an additional assay to confirm the altered tension at junctional area: (see also response to referee #1). This technique independently assesses changes in cell-cell junctional contractility by measuring the recoil of labeled plasma membrane from the cut site. The data show that initial recoil of the E-cadherin-labeled junctions is significantly slower in cells with Dsg1-FL compared to mCherry control or Dsg1-909 mutant expressing cells signifying a decrease in contractile tension. **We included this data to Figure 6i-k and Supplementary Video 1-3.**

We also performed cell-ECM traction force microscopy to assess alterations in the distribution of cell-substrate traction stresses within a colony, which have been shown to depend on the presence of functional adherens junctions. The measurement of cell-ECM traction forces revealed that Dsg1-FL colonies are less contractile than colonies expressing the Dsg1-909 mutant. Moreover knockdown of Tctex-1 in cells expressing endogenous Dsg1 increased staining of phosphorylated myosin light chain (p-MLC) further confirming an increase in actin contractility at cell-cell interface when Dsg1-Tctex-1 interactions are impaired (the new data included in **Supplemental Figure 5**). Overall these data suggest that dynein-Tctex-1-dependent Dsg1 positioning at cell junctions reduces actomyosin contractility in cells and the change in actomyosin contractility affects tension distribution at the junctions.

Comment: Although it is not quantified, the knockdown of Dsg1 in the stratification experiments shown in Fig 6 looks extremely efficient – there is none in the knockdown - yet the accompanying change in stratification is relatively small. How can this be if we are really dealing with the mechanism that regulates delamination?

Response: The referee is correct that the knockdown for Dsg1 is quite efficient, raising the question as to whether the observed changes in delamination are causally related to loss of Dsg1. As referee #1 pointed out, a limitation of this assay was the absence of a Day 0 (D0) time point and thus there was no “starting point” for comparison to the Day 1 (D1) knockdown cultures, formally making it difficult to rule out other explanations for our data. To address this concern, we added a D0 baseline experiment to calculate the number of single superficial cells present in the population at first plating, in order to be able to essentially subtract out that component. Specifically, we compared D0 and D1 (24h of stratification) rafts, assessing the percentage of single basal cells, symmetric divisions, asymmetric divisions and delamination. Our data showed that after cell seeding we detected an average of 21% of single GFP labeled cells on the top of the monolayer, 19% of symmetric divisions, 4% of asymmetric divisions and 56% of single basal cells. These data demonstrate that at the start point not all GFP labeled cells are attached to the collagen plug. However, after 24 hours of stratification (D1 rafts) the amount of delamination events significantly changed: they are increased up to 44%, while the amount of single basal cells decreased to 35%.

At D1, Dsg1 knockdown decreased delamination events from 48% to 37% and Tctex-1 knockdown decreased delamination events from 46% to 29%. For the purpose of comparison, if one were to subtract out the 21% of pre-existing suprabasal cells at D0, it gets 29% siNeg and 16% siDsg1 delamination events, an almost 2-fold difference. While knockdown of Dsg1 does not totally abrogate delamination events, these data provide compelling support for the importance of Dsg1 promoting efficient delamination in this model system. The

more dramatic impact of Tctex-1 knockdown raises the possibility that in the absence of Dsg1, Tctex-1 may be able to couple to other machinery to help compensate for the loss of Dsg1.

Minor points

Comment: The plural 'media' is used almost throughout when the intention seems to be the singular 'medium'.

Response: Thank you for this suggestion, the "media" word was replaced with "medium" though out the text.

Comment: Line 114: Since the authors have not actually defined a Tctex-1 binding site, would 'binding region' be a better term here?

Response: Thank you for the suggestion; this has been changed in the text.

Comment: Line 159: Surely 'different.....from' rather than 'differentthan'.

Response: This has been corrected.

Comment: The pictures of mouse embryos in Fig 6 are superfluous.

Response: The pictures were removed.

Comprehensive Outline of changes for Editor and both Referees:

Data that explore in more detail the specificity of Tctex-1-desmoglein 1 interactions and show that Tctex-1 does not associate with E-cadherin and Dsg2- positive junctions.

New Figure 2a: New panel a (replaced former panel c in Figure 1) showing accumulation of Tctex-1 with Dsg1-FL but not with Dsg2 or E-cadherin.

New Figure 2b: New panel b showing quantification of Pearson correlation for Dsg1-FL/Tctex-1, Dsg2/Tctex-1 and E-cadherin/Tctex-1 respectively.

New Figure 2c: New panel c is a proximity ligation assay (PLA) showing association of Tctex-1 and Dsg1-FL but not Dsg2 or E-cadherin in situ.

New Figure 2d: New panel d showing quantification of PLA for Dsg1-FL-Tctex-1, Dsg2-Tctex-1 and Ecad-Tctex-1 pair with or without Tctex-1 knockdown.

New Figure 2e: Immunoblot of Tctex-1 siRNA knockdown corresponding to PLA experiments for Dsg1-FL-Tctex-1, Dsg2-Tctex-1 and Ecad-Tctex-1 pairs.

New Figure 2f: New panel f showing distribution of Dsg1 and Tctex-1 in human skin.

New Figure 2g: New panel g showing structured illumination microscopy of Dsg1 and Tctex-1 in basal keratinocytes of human skin.

New Figure 3a (former Figure 2b): Added panels of immunofluorescence showing Dsg2 and E-cadherin distribution at the cell borders with or without Tctex-1 knockdown.

New Figure 3b: New panel b showing line scan analysis of cadherins intensity at the cell-cell interface with or without Tctex-1 knockdown.

New Figure 3d: New panel d showing line scan analysis of ectopic Dsg1 proteins at the cell borders.

New Figure 3e (Former Figure 2f): Added immunoblot with E-cadherin from subcellular fractionation of keratinocytes, showing that Dsg1-909 predominantly accumulates in the same fraction as E-cadherin.

New Figure 5a (Former Figure 4a): Added immunoblot with E-cadherin showing that it is not associated with Dsg1-cortactin complex.

Supplemental Fig 2: New panel c is immunoblots showing levels of ectopic proteins expression (top) and levels of Tctex-1 and DP knockdown (bottom). New panel d shows that abrogation of Dsg1-Tctex-1 interactions do not affect adhesion strength within the keratinocyte monolayer (quantifications are included).

Supplemental Fig 6: New panel c showing knockdown of Dsg1 in differentiated NHEKs abolished cortactin accumulation at the junctional area.

Additional data show changes in cell junctional tension to complement observations made by FRET and vinculin staining.

New Figure 6 (Former Figure 5): New panel i showing expression of Dsg1-FL, Dsg1-909 and E-cad-YFP in laser ablation experiments.

New Figure 6 (Former Figure 5): New panel j showing live cell imaging analysis of contractile tension at E-cad-YFP in cells expressing mCherry, Dsg1-FL or Dsg1-909. The figure includes panels from video before and after laser ablation at the cell junctions.

New Figure 6 (Former Figure 5): New panel k is quantification of recoil curves of apical adherens junctions in mCherry-control, Dsg1-FL or Dsg1-909 expressing cells.

New Supplemental Fig 5: New panel a showing distribution of strain energy density for cell colonies expressing Dsg1-FL or Dsg1-909 mutant. New panel b: quantification of strain energy per area for colonies expressing Dsg1-FL or Dsg1-909 mutant. New panel c and d showing decrease in phosphorylated myosin light chain staining at the junctional area in differentiated NHEKs with Tctex-1 knockdown.

New Supplemental Videos 1-3: Video of NHEKs expressing Ecad-YFP with Dsg1-FL, Dsg1-909 or mCherry before and after laser ablation at the E-cadherin-labeled junctions.

Additional experimental analysis supporting the idea that desmoglein 1 promotes stratification via delamination rather than asymmetric cell division.

New Figure 7: Panel a is X-Z Apotome images of MDCK monolayers expressing GFP or ectopic Dsg1 proteins showing that Dsg1-FL expression promotes extrusion of MDCK cells.

New Figure 7: New panel b is a quantification of amount of suprabasal cells on the top of the MDCK monolayer expressing GFP, Dsg1-FL or Dsg1-909 and an immunoblot showing level of ectopic Dsg1 protein expression.

New Figure 7: New panel c is a quantification of the amount of suprabasal cells on the top of the MDCK monolayer expressing ectopic Dsg1FL with or without Tctex-1 knockdown and an immunoblot showing level of Tctex-1 knockdown.

Figure 8 (Former Figure 6): New panel c showing the comparison of population analysis between starting point (Day 0) and Day 1 of organotypic raft development.

New Supplemental Fig 8: New panels a and b showing that expression of ectopic Dsg1 proteins does not induce proliferation in MDCK cells. New panels c and d showing that expression of ectopic Dsg1 proteins does not induce apoptosis in MDCK cells.

New Supplemental Fig. 9: New panels a and b showing that knockdowns of Dsg1 or Tctex-1 do not prevent centrosomes repositioning to the apical side in basal layer of D1 organotypic raft cultures. New panels c and d showing distribution of division axis for telophase cells in the basal layer of D1 raft cultures for siCtrl, siDsg1 and siTctex-1 conditions.

New Supplemental Video 4 and 5: Video of cell extrusion from MDCK monolayer expressing Dsg1-GFP versus GFP.

Revised Model

Figure 9 (Former Figure 7): Revised model includes insoluble membrane compartments at the plasma membrane as well as sequential steps that promote stratification out of basal layer.

We have also included the following data in this response for the referees' benefit:

- Co-IPs showing that E-cadherin does not associate with Tctex-1 when Dsg1 is expressed.
- Lipid raft isolations from NHEKs lysates with or without Tctex-1 knockdown.

Reviewers' comments:

Reviewer #1 (Remarks to the Author):

The authors have made substantial changes to the document and included important new data, including that Dsg1 is sufficient to induce delamination in simple epithelia. While improved, there are still a number of concerns that should be addressed before publication. Most of these should be doable with text changes or with analysis of existing data.

Major points

1. The 3D culture data on delamination are currently still quite weak and require additional attention. I have two major concerns – first the data seem to suggest that there is little or no cell division occurring in this context. Between day 0 and day 1 there is no increase in the total number of divided cells - symmetric and asymmetric divisions (23% total at D0 and 21% at D1). The other concern is the consistency of the findings and their significance. Delamination percentages vary between 33 and 44% in the control samples. Yet Dsg1 knockdown (37% delamination) is reported to affect this process while Dsg2 knockdown (37% delamination) is said to have no effect. More data must be provided on the reproducibility and the statistical significance of these findings.

2. The authors should be more explicit about their model in the discussion (and in abstract and title). I think at present it would be less controversial to say that Dsg1 promotes delamination rather than stratification. When stratification is discussed, it usually is referring to the production of spinous/granular cells, not periderm. This should be stated more explicitly to prevent confusion.

3. Tctex-1 co-localization is shown with overexpressed Dsg1, and as controls, endogenous E-cadherin and Dsg2 are shown. Can co-localization with endogenous be seen? If the other adhesion molecules are overexpressed, is co-localization seen? The current comparison is not a useful control if the same conditions aren't used for the other molecules.

Overexpression or knockdown are used throughout the paper for different experiments and it is not always clear why the given approach was used. A sentence or two on rationale for these choices would be appreciated.

In addition, the co-localization data in intact skin is not convincing (Fig 2g). A Pearson's quantitation of co-localization (with an E-cadherin control) would significantly strengthen this.

Minor

4. The authors should address why other desmosomal and adherens junction components organization looks normal while the specific organizations of Dsg1, desmoplakin and F-actin are disrupted. I would have expected that all desmosome components are affected in desmoplakin and Dsg1 are altered. For example, are there two clear different types of desmosomes in these cells, Dsg1 positive versus Dsg2 positive?

5. Please include quantitation of difference in colocalization with gangliosides for Dsg1-FL and Dsg1-909.

6. Some figures need reorganization for clarity, such as Figure 6.

7. The mechanistic connection between Dsg1 and Tctex-1 and cortactin remain weak.

Reviewer #2 (Remarks to the Author):

Firstly, let me congratulate the authors on a substantial amount of additional work and an extremely thorough response to the reviewers' comments. As before, I think that the manuscript contains many fascinating observations indicating a role for Dsg1 and its association with Tctex-

1 and cortactin/Arp2/3 in remodelling junctional actin and reducing adherens junction tension, leading to epidermal stratification. However, for me there still remains a certain amount of confusion and a new issue has arisen. In order to be brief I will try to focus on these points.

The addition of immunofluorescent staining for Dsg1 and Tctex-1 in human epidermis is most welcome. However, it raises a problem. It seems that almost all of the basal cells exhibit peripheral Tctex-1 staining and, although the staining is not quite so clear, all or most of them also exhibit Dsg1 staining. The authors suggest that expression of Dsg1 is sufficient for stratification on the basis of their fascinating new result showing that Dsg1 expression in MDCK cells causes them to stratify. If that is the case, why do not all of the basal epidermal cells stratify, because all or most of them express Dsg1? Clearly they do not because that would be disastrous for the epidermis, but why don't they?

The authors have provided considerable new data in relation to the section on targeting of Dsg1 to the insoluble membrane pool by Tctex-1, but the data now seem even more confusing. Fig.3 a,b shows that knockdown of Tctex-1 alters the distribution of Dsg1 at the cell membrane but does not alter Dsg2 and E-cad distribution. Yet the majority of both the Dsg1 and Dsg2 must be in desmosomes. So are they in different desmosomes? It seems more likely that they are in the same desmosomes as there is much precedence for different desmosomal cadherins occurring together in the same desmosomes in the literature, and if that is the case, how can their distributions differ? And does this mean that the distribution of desmosomes is dependent on Tctex-1 once Dsg1 is expressed? And does this mean that the distribution of Dsg1-expressing desmosomes is regulated by actin rather than keratin? Furthermore, the observation on E-cad needs a control: what is its distribution in the presence of Tctex-1? This is crucial because Dsg1 and Tctex-1 are shown later to alter adherens junction tension. Do they do this without altering AJ distribution? Lastly, I'm afraid I don't understand the sentence on line 152, beginning "A similar distribution ..." Similar to what?

Reviewer # 1 (Remarks to the Author):

The authors have made substantial changes to the document and included important new data, including that Dsg1 is sufficient to induce delamination in simple epithelia. While improved, there are still a number of concerns that should be addressed before publication. Most of these should be doable with text changes or with analysis of existing data.

Major points

Comment : 1. The 3D culture data on delamination are currently still quite weak and require additional attention. I have two major concerns – first the data seem to suggest that there is little or no cell division occurring in this context. Between day 0 and day 1 there is no increase in the total number of divided cells - symmetric and asymmetric divisions (23% total at D0 and 21% at D1). The other concern is the consistency of the findings and their significance. Delamination percentages vary between 33 and 44% in the control samples. Yet Dsg1 knockdown (37% delamination) is reported to affect this process while Dsg2 knockdown (37% delamination) is said to have no effect. More data must be provided on the reproducibility and the statistical significance of these findings.

Response: The reviewer raised two concerns regarding the 3D epidermal organotypic cultures experiments. The first comment relates to the extent to which **cell division** is occurring in these experiments, stating that “between day 0 and day 1 there is no increase in the total number of divided cells”. In fact, we have calculated based on growth curves that at every time point in the experiment cells are actively dividing, at rates approaching 25-30% each day. Confusion may have arisen because we were not clear enough about the experimental design. In these experiments, cells are mixed at a ratio of 10% GFP labeled cells to unlabeled cells, followed by treatment with the designated RNAi or control, and after 48 hours either harvested to determine the baseline of events at the time of lifting (Day 0) or lifted to an air medium interface to stimulate stratification and then harvested at 24 hrs (day 1). Then we count 4 types of event (not cells), represented by single suprabasal cells (putative delamination event); a pair of basal cells (SD); a pair of cells in which one is basal and one suprabasal (AD); and a single basal cell. Because both green cells and unlabeled cells are dividing, the total percentage of green cells that had undergone either an AD or SD at each time point should be the same proportion of all cells (labelled and unlabeled), and the 23% and 21% (AD plus SD) is about what would be expected based on our growth curve analysis. *The experimental design is clarified starting at line 356 of the text. We also revised the caption for Fig. 8 to specify this time point as time of lifting not time of first plating. Note it is likely that a proportion of the suprabasal single cells counted at Day 0 are in fact cells that delaminated in submerged culture prior to air-lifting, but considering the day of lift as the baseline is the most conservative approach.*

To address the second concern regarding the significance of change for the percentage of delamination events in the Tctex-1, Dsg1 and Dsg2 knockdown conditions in comparison to the control, we carried out additional statistical analysis as suggested by the referee. It is important to note that each experiment was performed in pairwise fashion such that the same human keratinocyte clone at the same passage number was used for the control and experimental knockdowns. As we observed substantial inter-clonal variability in this assay, we show the data as pairwise comparisons. (Note to reviewer, this inter-clonal variability likely explains at least in part differences among control populations when comparing experiments done with different clones and at different times). To show reproducibility of these pairwise comparisons we plotted the data in two additional ways. In **new Supplementary Figure 10a** we show the paired differences (each difference is represented by a single point) between si control and si experimental groups for each of the four types of event we tracked. By representing these as paired differences, we can now compare across the different experimental arms (Tctex KD, Dsg1 KD, and Dsg2 KD) for each type of event. A positive difference represents a decrease compared with control, whereas a negative difference represents an increase. Using a paired two-tailed t test, Tctex-1 ($p=0.0446$) and Dsg1 ($p=0.0238$) knockdowns consistently exhibited an increase in the percentage of single basal cell events in each experiment, while Dsg2 knockdown experiments did not reveal the same trend ($p=0.7997$). In the case of

single suprabasal cell events, Tctex-1 ($p=0.0083$) and Dsg1 ($p=0.0288$) consistently exhibited a decrease in each experiment, while Dsg2 knockdown experiments did not reveal the same trend ($p=0.3189$). We carried out a similar pairwise analysis for percentage of SD and AD events. While our analysis failed to detect a

Figure 1 for referees: String plots showing pairwise comparisons for siTctex, siDsg1 and siDsg2 for each of 4 “events” described in text.

significant difference or trend under all experimental conditions, because the relative number of events in the analysis of divisions was small compared to the percent single basal or suprabasal cells, we have now exerted more caution in the text, so as not to rule out a difference that might have been observed were we to look at a larger number of events using this assay (see lines 381-382, 392-394). To address this question in another way, we also included data (at the time of the last revision) on centrosome localization and spindle orientation using γ -tubulin or survivin analysis (**Supplementary Figure 9**). These strategies also did not reveal any alterations that would indicate a role for Dsg1:Tctex in controlling spindle orientations, under the conditions of our experiments. Nevertheless, we use cautionary language in the text stating that we do not rule out the possibility that Dsg1 could have an impact on this process in certain contexts (see lines 392-394).

Finally, we also plotted the data as string plots using the same data described above for pairwise comparisons (Fig. 1 for referees). These data are shown here for the referees, but not included in the text, as we felt that the paired difference plots were easier to digest (**Supplementary Figure 10a**). Altogether these data provide compelling support for a role for Dsg1 and Tctex in promoting single cell delamination.

Comment: 2. The authors should be more explicit about their model in the discussion (and in abstract and title). I think at present it would be less controversial to say that Dsg1 promotes delamination rather than stratification. When stratification is discussed, it usually is referring to the production of spinous/granular cells, not periderm. This should be stated more explicitly to prevent confusion.

Response: Thank you for this suggestion; we made the changes to the title and text (see lines 398, 400-401, 421, 456).

Comment: 3. Tctex-1 co-localization is shown with overexpressed Dsg1, and as controls, endogenous E-cadherin and Dsg2 are shown. Can co-localization with endogenous be seen? If the other adhesion molecules are overexpressed, is co-localization seen? The current comparison is not a useful control if the same conditions aren’t used for the other molecules. Overexpression or knockdown are used throughout the paper for different experiments and it is not always clear why the given approach was used. A sentence or two on rationale for these choices would be appreciated. In addition, the co-localization data in intact skin is not convincing (Fig 2g). A Pearson’s quantitation of co-localization (with an E-cadherin control) would significantly strengthen this.

Response:

Ectopic expression versus endogenous: The reviewer asked us to explain why we use ectopic Dsg1 expression in many of our experiments. Dsg2 and E-cadherin are constitutively expressed in keratinocytes, whereas Dsg1 expression is differentiation-dependent. Dsg1 expression is induced over a 2-3 day time period by growing keratinocytes in 1.2mM Ca²⁺ containing media. At this point, cells have begun to stratify and expression of endogenous Dsg1 is heterogeneous (mosaic) and graded, such that superficial cells express the most Dsg1. This can complicate imaging analysis and reduces the efficiency of PLA analysis (which works best in sub-confluent areas). We previously showed that introducing physiologically relevant levels (not over-expression) of Dsg1 into undifferentiated cultures promotes the biochemical program of differentiation (which is prevented by Dsg1 silencing), and in the current manuscript we show it can also stimulate changes in actin distribution that occur during differentiation (again, these changes are prevented by Dsg1 silencing). Thus, by precociously introducing physiologically relevant levels of Dsg1 we are initiating and synchronizing Dsg1-mediated processes, simplifying the analysis and allowing a comparison of wild type and mutant versions of Dsg1, without the complicating factor of having to first silence endogenous Dsg1. *As suggested, we clarified in the text when and why ectopic Dsg1 was used (see lines 133-138).*

Localization of Tctex and Dsg1: We carried out additional high resolution immunofluorescence analysis on both submerged cultures and in tissues to assess Tctex localization with respect to endogenous Dsg1 compared with E-cadherin (see new Fig. 1b,d). Tctex-1 was broadly distributed in both cases, and areas of Dsg1-Tctex colocalization were observed near the cell periphery. However, Pearson's co-efficients for Dsg1-Tctex-1 (0.07 ± 0.01 for submerged and 0.15 ± 0.01 for tissue) and E-cad-Tctex-1 (0.09 ± 0.02 for submerged and 0.18 ± 0.01 for tissue) are not appreciably different from each other. For this reason we carried out proximity ligation analysis, which provides a more informative strategy to complement the biochemical identification of the Dsg1-Tctex interaction and assess the specificity of Dsg1-Tctex proximity compared with other cadherins. We included a new set of PLA experiments in Figure 1 to compare the extent to which endogenous Dsg1 versus E-cadherin interact with Tctex. Both endogenous and ectopic Dsg1 but not E-cadherin (Fig. 1e-j) or Dsg2 (Fig. 1h-j), exhibit close association with Tctex. We also note that Tctex does not exhibit the sort of concentrated junctional localization expected for a stable structural component of the desmosome. This is consistent with the model

shown in Figure 9, depicting dynein/Tctex functioning in the targeted delivery of Dsg1 vesicles to the proper location at the membrane, rather than being a stable junctional component. Because Fig.2a in the previous version of the manuscript was confusing we have re-organized the figures to focus Figure 1 on the proximity analysis and localization of endogenous Tctex and Dsg1 (versus E-cad) in submerged cultures and intact epidermis.

Figure 2 for referees: Proximity ligation assay using primary antibody directed against Dsg1 and Tctex-1, Dsg2 and Tctex-1, E-cadherin and Tctex-1 was performed on undifferentiated NHEKs expressing Dsg1-mCherry, Dsg2-GFP or E-cadherin-mCherry respectively. The pair of IgG and Tctex-1 was used as a control. Blue DAPI staining marks nuclei. Quantification of PLA signal/field for each condition is shown below (mean \pm SEM, **p<0.01, unpaired two-tailed *t* test).

“Overexpression of other adhesion molecules”: In response to the reviewer’s question about whether overexpression of Dsg2 and E-cadherin will promote their interaction with Tctex-1, we performed proximity ligation assays for Tctex-1 with Dsg1, Dsg2 or E-cadherin in undifferentiated keratinocytes, which expressed Dsg1-mCherry, Dsg2-GFP or E-cadherin-mCherry. The results support the conclusion that Tctex-1 specifically associates with Dsg1 and not E-cadherin or Dsg2 even when those proteins are ectopically expressed in cells (**Figure 2 for referees**).

Minor

Comment: 4. The authors should address why other desmosomal and adherens junction components organization looks normal while the specific organizations of Dsg1, desmoplakin and F-actin are disrupted. I would have expected that all desmosome components are affected in desmoplakin and Dsg1 are altered. For example, are there two clear different types of desmosomes in these cells, Dsg1 positive versus Dsg2 positive?

Response: The referee raises an important point. To address whether there are desmosomes with different compositions in these cells, we carried out immunofluorescence co-localization of both ectopic and endogenous Dsg1 with Dsg2 and DP and have included these in this letter for the referee's information. Analysis of endogenous staining patterns along the Z plane (using the Zeiss Apotome) reveals most Dsg2 staining in basal to intermediate layers and most Dsg1 staining in intermediate to suprabasal layers of confluent differentiating cultures. This distribution is not surprising based on the distribution in intact tissue. If we look in the transitional region that has both endogenous Dsg1-positive desmosomes and Dsg2-positive desmosomes in these differentiating cultures (or, in the case of ectopically expressed Dsg1 in flatter cultures) there is substantial co-localization between Dsg1 and 2, but also areas where there is primarily Dsg1 or Dsg2 (see **Figure 3 for referees**). These data suggest that desmosomes with a range of Dsg1:Dsg2 ratios exist.

As we show in **new Supplemental Figure 2f**, Dsg1 and Dsg2 both co-localize with DP consistently, and in control KD cultures there are many areas where all three-- Dsg1, 2 and DP -- overlap. Interestingly, knockdown of Tctex-1 decreases Dsg1- and Dsg2- overlap at the cell-cell junctional interface in keratinocytes (**Supplemental Figure 2f**). While both pools of Dsg1 and 2 maintain their co-localization with DP, the Dsg1:DP staining is aberrant, consistent with the idea that Tctex association is necessary to deliver Dsg1, but not Dsg2, to the right place on the plasma membrane. DP co-localization with both Dsgs is at a somewhat reduced level when determining the Pearson's co-efficient, which is not surprising as the pool of DP is split between the two (Supplementary Figure 2f right panels). These data are consistent with the idea that Tctex is required for properly localizing Dsg1 at the plasma membrane, and its relationship relative to both Dsg2 and Ecadherin is shifted spatially in the absence of Tctex or the Tctex binding site on Dsg1.

Comment: 5. Please include quantitation of difference in colocalization with gangliosides for Dsg1-FL and Dsg1-909.

Response: The quantification of co-localization of ganglioside lipids with Dsg1-FL or Dsg1-909 is shown in **Figure 2 panel h**.

Comment: 6. Some figures need reorganization for clarity, such as **Figure 6**.

Response: To improve the presentation, we split **Figure 6** into two separate figures: **Figure 5** and **Figure 6** and re-organized them to keep related panels as close as possible.

Fig 3 for referees. Relative localization of endogenous and ectopic (FL-Dsg1) and Dsg2. Images of endogenous Dsg1 in the top panels were taken in the transitional zone where Dsg2 and Dsg1 are both expressed. Note that both endogenous and ectopic Dsg1 exhibit extensive overlap with Dsg2, but co-localization is not uniform.

Comment: 7. The mechanistic connection between Dsg1 and Tctex-1 and cortactin remain weak.

Response:

While we agree that there are still unanswered questions regarding the newly described Dsg1:Tctex protein complex, we feel strongly that the work makes an important advance in understanding a previously unappreciated function of Dsg1 and the molecular machinery that mediate this function. We showed that Dsg1 binds both Tctex-1 and cortactin and all three proteins are required to promote remodeling of actin cytoskeleton at the desmosomes in keratinocytes. While Tctex-1 and cortactin bind to Dsg1 independently, the Dsg1-Tctex-1 interaction is required to bring the Dsg1-cortactin complex to the insoluble membrane pool at the cell-cell interface, which in turn, leads to Arp2/3-dependent actin polymerization at Dsg1-positive desmosomes. The consequent re-organization of the cytoskeleton and changes in tension confer properties on cells that are important for morphogenesis.

Reviewer #2 (Remarks to the Author):

Firstly, let me congratulate the authors on a substantial amount of additional work and an extremely thorough response to the reviewers' comments. As before, I think that the manuscript contains many fascinating observations indicating a role for Dsg1 and its association with Tctex-1 and cortactin/Arp2/3 in remodelling junctional actin and reducing adherens junction tension, leading to epidermal stratification. However, for me there still remains a certain amount of confusion and a new issue has arisen. In order to be brief I will try to focus on these points.

Comment: The addition of immunofluorescent staining for Dsg1 and Tctex-1 in human epidermis is most welcome. However, it raises a problem. It seems that almost all of the basal cells exhibit peripheral Tctex-

1 staining and, although the staining is not quite so clear, all or most of them also exhibit Dsg1 staining. The authors suggest that expression of Dsg1 is sufficient for stratification on the basis of their fascinating new result showing that Dsg1 expression in MDCK cells causes them to stratify. If that is the case, why do not all of the basal epidermal cells stratify, because all or most of them express Dsg1? Clearly they do not because that would be disastrous for the epidermis, but why don't they?

Figure 4 for referees: Additional examples of X-Z scanned images of polarized MDCK cell monolayers expressing Dsg1-FL-Flag. Monolayers were stained for Flag (green), F-actin (red) and DAPI (marks nuclei). Dsg1-FL positive cells expressed in a mosaic pattern were extruded from the monolayer.

Response: The reviewer brings up an important point. Based on our collective observations we believe the explanation lies (at least in part) in the variable expression of Dsg1 within the basal layer or monolayer (depending on the context), and that there is a threshold level of Dsg1 required to mediate sufficient changes in tension necessary to promote delamination. This idea is in line with a report that altered tension at adherens junctions, driven by mosaic reorganization of F-actin along the junctions in single cells promotes cell extrusion from simple epithelia (Wu et al., Nat Cell Biol 2014, 16:167-178). Supporting this idea, in the MDCK cell experiments Dsg1 expression in the MDCK cells monolayer was always mosaic. Our observation that extruded Dsg1-expressing MDCK cells were surrounded by parental cells, supports this idea. Here we provide additional examples of mosaic expression of Dsg1 in MDCK cells that leads to cell extrusion (**Figure 4 for referees**). Along these same lines, from looking at examples of Dsg1 in many human samples, the expression of Dsg1 in basal layer can vary locally, which would be

expected to create a “mosaic” pattern of actin reorganization within single cells. Mosaic changes in actin, in turn, would be expected to alter the pattern of junctional contractility within the basal layer to allow single cell delamination into the suprabasal layer.

Comment: The authors have provided considerable new data in relation to the section on targeting of Dsg1 to the insoluble membrane pool by Tctex-1, but the data now seem even more confusing. Fig.3 a,b shows that knockdown of Tctex-1 alters the distribution of Dsg1 at the cell membrane but does not alter Dsg2 and E-cad distribution. Yet the majority of both the Dsg1 and Dsg2 must be in desmosomes. So are they in different desmosomes? It seems more likely that they are in the same desmosomes as there is much precedence for different desmosomal cadherins occurring together in the same desmosomes in the literature, and if that is the case, how can their distributions differ? And does this mean that the distribution of desmosomes is dependent on Tctex-1 once Dsg1 is expressed? And does this mean that the distribution of Dsg1-expressing desmosomes is regulated by actin rather than keratin?

Response: Referee #1 had a similar question, regarding whether Dsg1 and 2 are in the same or different desmosomes. We repeat that explanation here for the referees’ convenience. To address whether there may be desmosomes with different compositions in these cells, we carried out co-localization of both ectopic and endogenous Dsg1 with Dsg2 and DP and have included these in this letter for the referee’s information (**Figure 3 for referees**). Analysis of endogenous staining patterns along the Z plane (using the Zeiss Apotome) reveals most Dsg2 staining in basal to intermediate layers and most Dsg1 staining in intermediate to suprabasal layers of confluent differentiating cultures. This distribution is not surprising based on the distribution in intact tissue. If we look in the transitional region that has both endogenous Dsg1-positive desmosomes and Dsg2-positive

desmosomes in these differentiating cultures (or, in the case of ectopically expressed Dsg1 in flatter cultures) there is substantial co-localization between Dsg1 and 2, but also areas where there is primarily Dsg1 or Dsg2 (**Figure 3 for referees**). These data suggest that desmosomes with a range of Dsg1:Dsg2 ratios exist.

Figure 5 for referees. Dsg1 localization is not dependent on an intact actin cytoskeleton. Differentiated NHEKs were treated with 25uM cytochalasin D for 1 hour to disrupt the actin cytoskeleton, and stained for endogenous Dsg1 and actin (phalloidin).

As we show in **new Supplemental Figure 2f**, Dsg1 and Dsg2 both co-localize with DP consistently, and in control KD cultures there are many areas where Dsg1, 2 and DP all overlap. Interestingly, knockdown of Tctex-1 decreases Dsg1- and Dsg2- overlap at the cell-cell junctional interface in keratinocytes (**Supplemental Figure 2f**). While both pools of Dsg1 and 2 maintain their co-localization the Dsg1:DP staining is aberrant, consistent with the idea that Tctex association is necessary to get Dsg1, but not Dsg2, to the right place on the plasma membrane. DP co-localization with both Dsgs is at a somewhat reduced level when determining

the Pearson’s co-efficient, which is not surprising as the pool of DP is split between the two. These data are consistent with the idea that Tctex is required for properly localizing Dsg1 at the plasma membrane, and its relationship relative to both Dsg2 and Ecadherin is shifted spatially in the absence of Tctex or the Tctex binding site on Dsg1. Altogether these data suggest that the distribution of Dsg1- but not Dsg2-positive desmosomes depends on Tctex-1 and the dynein motor complex. What targets Dsg2 to the insoluble membrane pool requires further investigation.

In addition, in response to referee #2’s question as to whether Dsg1-expressing desmosomes are regulated by actin rather than keratin, we treated differentiated keratinocytes with cytochalasin D and Dsg1-positive

Figure 6 for referees. NHEKs mosaically expressing Dsg1-FL were analyzed for E-cadherin localization. Line scan analysis of border intensities for E-cadherin in cells with or without Dsg1-FL shown in the graph (at least 30 borders were analyzed per condition in representative experiment, $p=0.64078$, unpaired two-tailed t test).

desmosomes were analyzed by utilizing structured illumination microscopy. Our data show that Dsg1-positive desmosomes were not affected by interfering with the actin cytoskeleton (**Figure 5 for referees**). This result, together with data provided in our manuscript, is consistent with the idea that while Dsg1 desmosomes are not dependent on the actin cytoskeleton, the proper localization of Dsg1-positive desmosomes promotes peri-junctional actin reorganization in keratinocytes.

Comment: Furthermore, the observation on E-cad needs a control: what is its distribution in the presence of Tctex-1? This is crucial because Dsg1 and Tctex-1 are shown later to alter adherens junction tension. Do they do this without altering AJ distribution?

Response: The E-cadherin distribution in the presence and absence of Tctex-1 is shown in **Figure 2a** and quantified in **Figure 2b**. We also would like to point out that we haven't detected any significant changes in E-cadherin distribution when Dsg1 full length or the Tctex-1 deficient mutant are expressed in keratinocytes (please see **Figure 2g**, panel with E-cadherin staining). To further confirm that Dsg1 presence in the cells does not alter E-cadherin distribution at the cell junctional area, we performed line scan analysis of E-cadherin in cells with mosaic Dsg1-FL expression and included the data in **Figure 6 for referees**.

Comment: Lastly, I'm afraid I don't understand the sentence on line 152, beginning "A similar distribution ..." Similar to what?

Response: Thank you for pointing this out; we made the changes to the text on line 166.

REVIEWERS' COMMENTS:

Reviewer #1 (Remarks to the Author):

Acceptable for publication.

Reviewer #2 (Remarks to the Author):

Once again the authors have provided more data and a very detailed response to the reviewers' comments. However, there are still some problems.

The epitopes for Dsg1 and Tctex-1 shown in Figs 1b and d are not co-localized so this should not be claimed in line 120, and I think it is not even correct to say that they show "areas of co-localization" (line 124). Compare, for example, with Supplementary Fig 2b where Dsg1 is truly co-localized with a known desmosomal component, DP. The PLA result is consistent with them being in close proximity, which, in turn, is consistent with the immunofluorescence. So the question arises as to whether they bind to each other in vivo. IP is not definitive on this point. If they are, why do they not co-localize?

There is also something I still find profoundly puzzling about the location of Dsg1 and the 909 mutant. Returning to SFig 2b, the co-localization of 909 with DP is much less strong than Dsg1, suggesting some of 909 is in desmosomes but much of it is not. This appears consistent with the solubility data in Fig 2 e, f, where much of the 909 appears in the soluble, non-desmosomal fraction with E-cadherin. (N.B. Fig 2e would be more informative if it included a western blot for DP as a desmosomal marker.) Even though much of it is not in desmosomes, 909 "increased adhesive strength comparably" with Dsg1. So adhesive strength does not depend on whether or not the protein is in desmosomes. Moreover, silencing Tctex-1 also did not affect cell-cell adhesion, even though this apparently alters the tension in adherens junctions. Which means that the level of tension in adherens junctions is not related to the strength of cell-cell adhesion, does it not? Further, silencing Tctex-1 sends most of the Dsg1 to the non-desmosomal fraction (SFig. 2 e), so the majority of it is now not in desmosomes, which seems to be rather contrary to what the authors claim. This makes it even more surprising that silencing Tctex-1 does not affect the strength of cell-cell adhesion. Perhaps the authors can explain this conundrum. And does this mean that variation in the strength of cell-cell adhesion is not important for stratification/delamination?

The epitopes for Dsg1 and Tctex-1 shown in Figs 1b and d are not co-localized so this should not be claimed in line 120, and I think it is not even correct to say that they show “areas of co-localization” (line 124). Compare, for example, with Supplementary Fig 2b where Dsg1 is truly co-localized with a known desmosomal component, DP. The PLA result is consistent with them being in close proximity, which, in turn, is consistent with the immunofluorescence. So the question arises as to whether they bind to each other in vivo. IP is not definitive on this point. If they are, why do they not co-localize?

Response:

In the paper we demonstrated by several complementary assays including yeast two hybrid, proximity ligation assay and co-immunoprecipitation, that Tctex and Dsg1 are in a biochemical complex and are closely associated in cells. The PLA analysis in particular is informative as it provides information about interactions in cells without extracting the cell contents or destroying the cell. In addition, an advantage of PLA is that it reduces “noise” of fluorescence from Tctex, which is known to be present throughout the cytoplasm (in dynein motor-associated and unassociated pools). Thus, PLA allows one to identify whether a specific protein is within proximity and thus likely to be a preferred cargo. Further, because PLA is a more sensitive assay (due to the exponential DNA amplification process) a small number of interactions can be more readily visualized within the broadly distributed patterns visualized by conventional immunofluorescence. This strategy was previously helpful to us, when we analyzed associations between another desmosomal cadherin pair (Dsg2 and Dsc2) and specific kinesin motors—minimal co-localization by fluorescence, but significant proximity by PLA which was shown to be functionally important in controlling the localization of these cadherins (Nekrasova, et al. *J. Cell Biol.* 195: 1185. 2011).

We agree with the referee that by immunofluorescence co-localization between Tctex and Dsg1 is not comparable to the robust co-localization exhibited by the stable core components of the desmosome (e.g. Dsg1 and DP). This is not surprising, based on Tctex’s broad distribution in the Golgi, Rab3D positive vesicular compartments, and association with various cargos in addition to Dsg1. Also, because of the dynamic nature of cargo-dynein complexes and the number of cargos dynein transports within the cells, only small percentage would be expected to show co-localization with a specific cargo at any one point in time.

In summary, while any one technique is not definitive, when taken together with the functional data, Y2H, PLA and co-IP all strongly support the conclusion that Tctex-1 as a part of the dynein motor associates preferentially with the extreme C-terminal domain of Dsg1 to control Dsg1 localization. *We have altered the text on lines 120, 122-123 to say “in close proximity” and have avoided the word “co-localization”.*

There is also something I still find profoundly puzzling about the location of Dsg1 and the 909 mutant. Returning to SFig 2b, the co-localization of 909 with DP is much less strong than Dsg1, suggesting some of 909 is in desmosomes but much of it is not. This appears consistent with the solubility data in Fig2 e, f, where much of the 909 appears in the soluble, non-desmosomal fraction with E-cadherin. (N.B. Fig 2e would be more informative if it included a western blot for DP as a desmosomal marker.) Even though much of it is not in desmosomes, 909 “increased adhesive strength comparably” with Dsg1. So adhesive strength does not depend on whether or not the protein is in desmosomes. Moreover, silencing Tctex-1 also did not affect cell-cell adhesion, even though this apparently alters the tension in adherens junctions. Which means that the level of tension in adherens junctions is not related to the strength of cell-cell adhesion, does it not? Further, silencing Tctex-1 sends most of the Dsg1 to the non-desmosomal fraction (SFig. 2 e), so the majority of it is now not in desmosomes, which seems to rather contrary to what the authors claim. This makes it even more surprising that silencing Tctex-1 does not affect the strength of cell-cell adhesion. Perhaps the authors can explain this conundrum. And does this mean that variation in the strength of cell-cell adhesion is not important for stratification/delamination?

Response

The referee asks why the co-localization of Dsg1-909 with DP is less strong than Dsg1 in SFig 2b. This is due to a phenomenon that we tried to explain in last round of review: under conditions where there is loss of Tctex-1 or Tctex-1 binding (i.e. the 909 mutant), Dsg1 becomes partially segregated from other desmosomal cadherins in the cell, such that the pool of DP, which does not change in size) is split between Dsg1 and other desmosomal cadherins (shown for Dsg2 under Tctex KD conditions in SF2f). Because of this phenomenon, the Pearson's co-efficients are reduced under both conditions (*now also shown for S2b in addition to 2f in revised Supplemental Figure 2*). But the important thing to keep in mind is that, collectively, the desmosomal cadherin-DP-IF connections remain the same. These data are consistent with the data in Supplementary Figure 4b, showing that cells expressing Dsg1-FL and Dsg1-909 have comparable intermediate filament attachments at cell-cell borders, thus demonstrating that the intermediate filament attachment function of desmosomes is not visibly perturbed. These observations are consistent with the results of the dispass assay, which is an assay that measures mechanical resistance to dissociation, for which intermediate filament-desmosome attachments are absolutely critical (Hudson, et al. *Methods. Cell Biol.* 78: 757. 2004). Thus, the force required to separate cells from each other in epithelial sheets expressing either Dsg1-WT or Dsg1-909 is comparable. *We have included further explanation of this phenomenon in the revised version of the text (lines 184-185) and caption to SFig2.*

With respect to solubility, the solubility of desmogleins is not the best indicator of its presence in desmosomes. In early work, it was shown that desmogleins become insoluble during their biosynthesis before they reach the plasma membrane (Pasdar, et al. *J. Cell Biol.* 113: 645. 1991. This shift in solubility occurs prior to association with desmoplakin or intermediate filaments. While this process of becoming insoluble is still poorly understood, more recently it has been reported that desmosomes (but not adherens junctions) are present in specialized lipid domains, thus affecting their biochemical compartmentalization and solubility. Our data suggest that Dsg1-Tctex-1 interactions properly position Dsg1-positive desmosomes in a specialized biochemical compartment (enriched in gangliosides) distinct from E-cadherin. Interference with Dsg1-Tctex interactions (e.g. via Dsg1-909 expression or Tctex-1 silencing) inhibits Dsg1's recruitment to this domain. However, our data show that this alteration in solubility does not preclude desmoglein from associating with desmoplakin, nor does it preclude desmoglein-associated desmoplakin from interacting with intermediate filaments to mediate strong adhesion, as shown in Suppl. Figs. 2,4. To be clear, we do not conclude from our data that "silencing Tctex-1 sends most of the Dsg1 to the non-desmosomal fraction". *We have clarified in the text that while Dsg1 becomes more soluble when interactions with Tctex are compromised, this does not detectably affect its association with desmoplakin, IF attachment or integrity of epithelial sheets (lines 177-178; 238-240).*

Finally, the referee asks "And does this mean that variation in the strength of cell-cell adhesion is not important for stratification/delamination?" No we would not conclude this. In fact, the onset of desmoglein expression in a mosaic fashion during epidermal stratification/delamination is expected to locally alter adhesion in cells destined to stratify. What we would propose, and the model that our data support, is that beyond its role in adhesion, Dsg1 is playing an additional role required for delamination. That is, through its association with Tctex, Dsg1 properly localizes the actin remodeling complex relative to E-cadherin, thus temporarily reducing cortical tension required for delamination. This idea is consistent with the conclusion of a paper from Sarah Wickstrom and colleagues that was published in NCB after submission of the revised version of our paper (Miroshnikova, et al, *Nat. Cell Biol.* 20: 69, 2018). Their data support a model predicting that keratinocytes with increased cell-cell adhesion force and decreased cortical tension will delaminate. Our data suggest that the Dsg1 expression is critical for mediating this decrease in cortical tension.

Stated another way, delamination requires a rapid rearrangement of cell shape and adhesions without disrupting mechanical integrity. Thus, it makes sense that we observe alterations in actin-based adhesions without grossly disturbing the IF system required for resisting external mechanical challenges. That the Tctex-1 deficient phenotype naturally sorts out these two modes and uses of adhesion machinery is a strength and main message of the paper. *We have clarified these points on page 15-16 (lines 451-472) of the text.*